# Adjuvant nivolumab, capecitabine or the combination in patients with residual triple-negative breast cancer: the OXEL randomized phase II study

Chemotherapy and immune checkpoint inhibitors have a role in the post-neoadjuvant setting in patients with triple-negative breast cancer (TNBC). However, the effects of nivolumab, a checkpoint inhibitor, capecitabine, or the combination in changing peripheral immunoscore (PIS) remains unclear. This open-label randomized phase II OXEL study (NCT03487666) aimed to assess the immunologic effects of nivolumab, capecitabine, or the combination in terms of the change in PIS (primary endpoint). Secondary endpoints included the presence of ctDNA, toxicity, clinical outcomes at 2-years and association of ctDNA and PIS with clinical outcomes. Forty-five women with TNBC and residual invasive disease after standard neoadjuvant chemotherapy were randomized to nivolumab, capecitabine, or the combination. Here we show that treatment with immunotherapy containing arms (nivolumab or a combination of nivolumab plus capecitabine) leads to an increase in PIS from baseline to week 6 compared with capecitabine alone, meeting the pre-specified primary endpoint. In addition, the presence of circulating tumor DNA (ctDNA) is associated with disease recurrence, with no new safety signals in the combination arm. Our results provide efficacy and safety data on this combination in TNBC and support further development of PIS and ctDNA analyses to identify patients at high risk of recurrence.

Breast cancer is the most commonly diagnosed cancer in women and the leading cause of cancer death in women worldwide[1]. Triple-negative breast cancer (TNBC) is an aggressive subtype that affects 10–15% of patients with breast cancer[2]. Compared to hormone receptor-positive (HR+) and human epidermal growth factor receptor 2 (HER2)-positive breast cancer, TNBC is often diagnosed in younger women and has higher rates of distant recurrence within 2–3 years of diagnosis[3,4]. For patients with metastatic TNBC, overall survival (OS) ranges from 10–23 months[5,6].

Most patients with early-stage TNBC are treated with neoadjuvant chemotherapy. Those who experience a pathologic complete response (pCR) have a significantly lower risk of recurrence and better survival outcomes than patients with residual invasive disease[7–12]. To reduce the risk of recurrence, patients with residual invasive disease after neoadjuvant chemotherapy are often treated with adjuvant capecitabine based on the results of the CREATE-X[13] and ECOG-ACRIN EA1131 trials[14]. However, patients with basal subtype TNBC treated with capecitabine in the post-neoadjuvant setting still experience only a 3-year invasive disease-free survival (iDFS) of 49% (95% confidence interval (CI), 39% to 59%)[14].

Aside from capecitabine, post-neoadjuvant treatment options for TNBC also include immune checkpoint inhibitors (ICIs) against

✉e-mail: Filipa_Lynce@dfci.harvard.edu

programmed cell death protein 1 (PD-1) and programmed death-ligand 1 (PD-L1). In the randomized phase III KEYNOTE-522 trial, patients with stage II-III TNBC were treated with neoadjuvant pembrolizumab or placebo in combination with chemotherapy. After surgery, patients on the neoadjuvant pembrolizumab combination arm continued to receive 1 year of adjuvant pembrolizumab, irrespective of response to neoadjuvant therapy. Patients who received pembrolizumab had superior pCR and event-free survival (EFS) rates compared to those on the placebo arm[15–17]. As a result, neoadjuvant pembrolizumab plus chemotherapy followed by 27 weeks of adjuvant pembrolizumab was approved by the U.S. Food and Drug Administration in 2021 for patients with high-risk early-stage TNBC.

Although KEYNOTE-522 did not allow the use of adjuvant capecitabine along with pembrolizumab, this combination is often utilized in clinical practice given the high risk of recurrence, the benefit observed with each drug alone, and the potential synergy when given together[18,19]. However, it is unclear if there is a benefit of PD-1/PD-L1 blockade in the adjuvant setting after preoperative chemotherapy without immunotherapy in patients without evidence of residual macroscopic tumor. Two trials, SWOG S1418[20] and A-BRAVE[21], are addressing this question.

In addition, it is crucial to identify mechanisms of response and resistance to ICIs to improve their efficacy and select patients who might derive the maximum benefit. Many investigations have focused on characterizing the tumor immune microenvironment[22,23]. However, this may be a challenge in the post-surgery setting when there is no evidence of macroscopic tumor. Thus, characterization of the peripheral immune system may provide potentially prognostic information about the effectiveness of immunotherapy.

## Table 1 | Landmark patient characteristics

| Characteristic | Total $N = 45$ | Arm A (nivo) $N = 15$ | Arm B (cape) $N = 15$ | Arm C (nivo + cape) $N = 15$ |
|---|---|---|---|---|
| Mean Age (SD) | 51.0 (11.5) | 46.3 (12.2) | 53.5 (8.8) | 53.1 (12.5) |
| Race | | | | |
| Black | 14 (31%) | 2 (13%) | 6 (40%) | 6 (40%) |
| White | 29 (65%) | 12 (80%) | 8 (53%) | 9 (60%) |
| Other | 2 (4%) | 1 (7%) | 1 (7%) | 0 |
| Ethnicity | | | | |
| Latino | 3 (7%) | 1 (7%) | 1 (7%) | 1 (7%) |
| Non-Latino | 42 (93%) | 14 (93%) | 14 (93%) | 14 (93%) |
| NACT | | | | |
| Taxane + anthracyclines | 42 (93%) | 14 (93%) | 15 (100%) | 13 (87%) |
| Taxanes only | 3 (7%) | 1 (7%) | 0 | 2 (13%) |
| Neoadjuvant carboplatin | | | | |
| Yes | 14 (31%) | 5 (33%) | 6 (40%) | 3 (20%) |
| No | 31 (69%) | 10 (67%) | 9 (60%) | 12 (80%) |
| Prior radiotherapy | | | | |
| Yes | 34 (76%) | 11 (73%) | 11 (73%) | 12 (80%) |
| No | 11 (24%) | 4 (27%) | 4 (27%) | 3 (20%) |
| Known germline mutation | | | | |
| BRCA1/2 | 3 (7%) | 2 (17%) | 1 (8%) | 0 |
| PALB2 | 2 (4%) | 0 | 2 (17%) | 0 |
| Pathological staging (yp) | | | | |
| I | 13 (29%) | 5 (33%) | 6 (40%) | 2 (13%) |
| II | 20 (44%) | 7 (47%) | 4 (27%) | 9 (60%) |
| III | 12 (27%) | 3 (20%) | 5 (33%) | 4 (27%) |

*NACT* neoadjuvant chemotherapy, *nivo* nivolumab, *cape* capecitabine.

Lastly, the detection of minimal residual disease (MRD) via plasma circulating tumor DNA (ctDNA) after the completion of neoadjuvant therapy has recently emerged as a strong predictor of recurrence and poor survival outcome[24–27]. Thus, prospective monitoring for ctDNA after the completion of neoadjuvant therapy might help to identify high-risk patients who could potentially benefit from intensified post-neoadjuvant salvage therapy leading to improved outcomes[28–32]. It may also offer a real-time approach to monitoring treatment efficacy.

In this work, we describe our investigator-initiated randomized phase II study (OXEL) of adjuvant nivolumab, capecitabine, or a combination of nivolumab and capecitabine in early-stage TNBC patients with residual invasive disease after the completion of neoadjuvant chemotherapy. The primary endpoint of the current study was to evaluate whether there were changes at 6 weeks vs. baseline in a peripheral immunoscore (immunoscore #1) that differed among patients who received immunotherapy (Arms A and C combined) compared to chemotherapy (Arm B). However, numerous studies published since the initiation of this study have also shown, in a range of solid tumors, that evaluation of landmark (pre therapy) levels of peripheral immune cell subsets has contributed the most valuable information in terms of immune correlates of clinical response. Landmark levels of ratios of neutrophils to lymphocytes[33–37], and frequencies of subsets of CD4$^+$ and CD8$^+$ T cells, B cells, monocytes, and natural killer (NK) cells[38–54] have been associated with clinical outcome in numerous solid tumors. Here, we also evaluated a second peripheral immunoscore (immunoscore #2) comprised of specific immune subsets only at landmark for association with disease recurrence, as an unplanned and exploratory endpoint. It should be noted that the immune cell components comprising the two peripheral immunoscores calculated in this study differ from each other, as the first evaluates the change of immune components as a result of different therapies, while the second analyzes the general immune status of patients prior to therapy. In this study, we also prospectively monitored ctDNA dynamics and investigated associations with recurrence and survival outcomes. Here we report the clinical and translational outcomes of the OXEL study.

## Results

### Patient and treatment characteristics

The OXEL study was an open label randomized phase II trial that enrolled patients with residual disease after neoadjuvant systemic therapy and surgery for TNBC. A total of 45 women were enrolled between August 2018 and June 2021, with 15 patients randomized to each arm. Patient characteristics are included in Table 1. The mean age was 51 years old. Most patients (93%) had been treated with neoadjuvant taxane plus anthracycline chemotherapy; 31% of patients had received carboplatin. Most patients (76%) had received prior adjuvant radiotherapy. Five patients (11%) had known germline pathogenic variants (3 in *BRCA1/2* and 2 in *PALB2*). Most patients had pathological stage (yp) II. There were no statistical differences in yp stage among the 3 arms ($p = 0.36$). The capecitabine dose intensity was similar for patients in Arms B and C and is included in Supplementary Table 1 and Supplementary Fig. 1. Of 45 patients, 42 underwent successful peripheral immune profiling by multicolor flow cytometry and 38 had primary tumor tissue that underwent successful whole exome sequencing (WES) with available ctDNA information; 35 patients underwent both immune profiling and ctDNA testing.

### Changes in a peripheral Immunocore (Primary Endpoint) and other immune cell subsets

Serial peripheral blood mononuclear cells (PBMCs) obtained from 42 patients (Arm A, $n = 15$; Arm B, $n = 14$; Arm C, $n = 13$) were evaluated before and after 6 and 12 weeks of therapy by multicolor flow cytometry for 158 immune cell subsets with defined biologic functions (Supplementary Table 2). We evaluated if the calculation of a peripheral immunoscore (immunoscore #1) could identify immunologic changes

that were unique to each treatment arm. Immunoscore #1 is calculated based on the frequency of specific immune subsets at landmark, 6 and 12 weeks, for which a biologic function has previously been reported[55–64] (Fig. 1a, Supplementary Table 3). Immunoscore #1, which is reflective of enhanced immune function, was not significantly altered after 6 weeks in patients enrolled in Arm A (who received nivolumab alone, $p = 0.217$), or Arm C (who received nivolumab plus chemotherapy, $p = 0.100$), but was significantly reduced ($p = 0.005$) in patients in Arm B who received chemotherapy alone (Fig. 1b). Significant increases in immunoscore #1 were noted at 6 weeks in analyses combining patients in Arms A and C (who received immunotherapy, $p = 0.040$). Only the reduction in the peripheral immunoscore in Arm B was maintained at 12 weeks (Supplementary Fig. 2A). A comparison of the % change in peripheral immunoscore #1 at 6 and 12 weeks vs landmark further highlights the statistical difference in the immunologic effects of chemotherapy and immunotherapy (Fig. 1c, Supplementary Fig. 2B).

Furthermore, additional distinct immune subsets showed statistical changes after 6 and 12 weeks of therapy that were specific to each treatment arm (Fig. 1d–f, Supplementary Tables 4–5)[65]. After 6 weeks, patients receiving capecitabine (Arm B) had decreases in proliferative CD8+ T cells (ki67+, $p = 0.019$), and increases in naïve CD4+ T cells ($p = 0.011$), CD8+ T cells that express CD73, a checkpoint involved in adenosine metabolism ($p = 0.020$), and NK cells that express the adhesion molecule CD226 ($p = 0.038$) (Fig. 1d). The reduction in ki67+ CD8+ T cells and increases in naïve CD4+ T cells and CD73+ CD8 + T cells were maintained at 12 weeks. In contrast, patients receiving nivolumab (Arm A) had a transient reduction at 6 weeks in double negative (DN, CD56dim CD16−) NK cells ($p = 0.025$) (Fig. 1e), a refined NK subset recently reported to be non-cytolytic and exhausted. Finally, patients receiving the combination therapy (Arm C) had transient increases at 6 weeks in conventional dendritic cells (cDC) ($p = 0.021$), a subset involved in antigen presentation, and proliferative effector memory (EM) CD4+ T cells that express ki67 ($p = 0.037$), while DN NK cells were increased after therapy both at 6 and 12 weeks ($p = 0.033$) (Fig. 1f).

### Survival and recurrence

At a median follow-up of 20.4 months, 16 patients (35.6%) had experienced a distant recurrence and 7 (15.6%) had died. Among all patients, the iDFS probability was 0.73 (+/− 0.07) at 1 year, 0.63 (+/− 0.08) at 2 years, and 0.54 (+/− 0.11) at 3 years (Supplementary Table 6). The median iDFS was longer for patients in Arm C compared to Arms A and B, with a 2-year iDFS in Arm C of 91% compared to 47% in Arm A or 53% in Arm B, although the difference did not reach statistical significance (Fig. 2a and Supplementary Table 6). The 2-year OS was longer for patients in Arms B and C, 80% and 83% respectively, compared to 70% in Arm A, but did not reach statistical significance (Fig. 2b).

### Toxicity

A total of 11 of the patients in the nivolumab group (Arm A), 14 in the capecitabine group (Arm B) and 14 in the combination group (Arm C) had at least one drug-related adverse event. There were 198 drug-related adverse events of any grade, with 29 (14.6%) reported in Arm A, 74 (37.4%) in Arm B, and 95 (48.0%) in Arm C. The most common drug-related adverse events in the safety analysis set are included in Table 2. Drug-related grade 3 toxicity was experienced by 7 patients (15.6%), with 2 (13.3%) in Arm B, and 5 (33%) in Arm C. There were no grade 4 toxicities or grade 5 adverse events (treatment-related deaths). There was no increase in immune related adverse events (irAEs) in Arm C. Only one patient (in Arm C) discontinued treatment due to drug-related adverse event.

### Peripheral immune subsets and disease recurrence

We next evaluated in exploratory post-hoc analyses whether there was any relation between the immune profile of patients prior to therapy and development of recurrence. In each treatment arm, distinct

immune subsets at landmark showed statistical association with the development of recurrence (Supplementary Table 7). In analyses combining all patients (Arms A, B and C), individuals who developed recurrence following treatment had lower levels of total natural killer T (NKT) cells ($p = 0.009$) and PD-1+ NKT cells ($p = 0.004$) at the landmark timepoint prior to therapy compared to patients who did not develop recurrence (Fig. 3a). In Arm A, patients with recurrence after nivolumab had higher landmark levels of naïve CD8+ T cells ($p = 0.021$) and regulatory T cells (Tregs) with a suppressive phenotype (HLA-DR+, $p = 0.040$), and lower levels of NK cells that express the activating receptor NKp30 ($p = 0.021$) and NKT cells that express PD-1 ($p = 0.021$) compared to patients not developing recurrence (Fig. 3b). In Arm B, patients developing recurrence after capecitabine had higher landmark levels of intermediate ($p = 0.042$) and non-classical ($p = 0.042$) monocytes compared to patients not developing recurrence (Fig. 3c). In Arm C, patients who recurred after the combination of capecitabine and nivolumab had appreciably higher landmark levels of total Tregs ($p = 0.026$) and Tregs with phenotypes each reflective of increased suppressive capacity (HLA-DR+, ICOS+, CD49d−) compared to individuals who did not recur (Fig. 3d).

While there were limited numbers of patients in each arm, distinct changes in specific immune subsets after 6 and/or 12 weeks of therapy were also associated with the development of recurrence (Supplementary Table 8). In Arm A, patients treated with nivolumab who developed recurrence had greater decreases after 6 weeks in double positive (DP, CD56brCD16+) NK cells ($p = 0.029$), an NK subset with both lytic and cytokine producing capabilities, and less of an increase after 12 weeks in naïve CD4+ T cells ($p = 0.021$) than patients who did not develop recurrence (Supplementary Fig. 3a). In Arm B, patients receiving capecitabine and developed a recurrence had similar immune profiles at 6 weeks compared to those patients who did not recur, but had greater increases in cDC ($p = 0.030$) and myeloid-derived suppressor cells (MDSC) ($p = 0.030$) after 12 weeks of therapy (Supplementary Fig. 3B). Patients enrolled in Arm C who developed recurrence after the combination of capecitabine and nivolumab had greater increases in terminally differentiated EMRA CD8+ T cells ($p = 0.026$) after 6 weeks of therapy, and greater increases in total CD8+ T cells ($p = 0.030$) and EMRA CD8+ T cells ($p = 0.030$) at 12 weeks compared to those patients who did not recur (Supplementary Fig. 3C).

To further interrogate the potential value of analyzing peripheral immune cell subsets, we conducted an unplanned post-hoc analysis to determine if the calculation of another peripheral immunoscore (immunoscore #2), would be of prognostic value in determining which patients in the current study may most likely benefit from therapy. Immunoscore #2 is based on specific refined immune cell subsets at landmark (Fig. 4a, Supplementary Table 9) for which a biologic function has been previously reported[56,66–73]. Immunoscore #2 was significantly associated with recurrence in patients receiving nivolumab alone (Arm A, $p = 0.005$) or nivolumab +/− capecitabine (Arms A and C combined, $p = 0.040$), but showed no association with recurrence in patients receiving capecitabine alone (Arm B, $p = 0.576$) (Fig. 4b). Moreover, patients in Arm A (receiving nivolumab, $p = 0.0003$) and Arms A and C combined (receiving nivolumab +/− capecitabine, $p = 0.0085$) with a landmark immunoscore #2 above the median (>11) had a longer iDFS compared to patients with an immunoscore #2 at or below the median in these arms (Fig. 4c). In contrast, landmark immunoscore #2 in Arm B (where patients received chemotherapy alone) was not associated with iDFS.

### Prior radiotherapy or *BRCA1/2* mutation status and landmark immune profile

We next investigated potential factors that may contribute to landmark variation in the immune profile of patients. We interrogated whether there were immunologic differences among patients based on

**a**

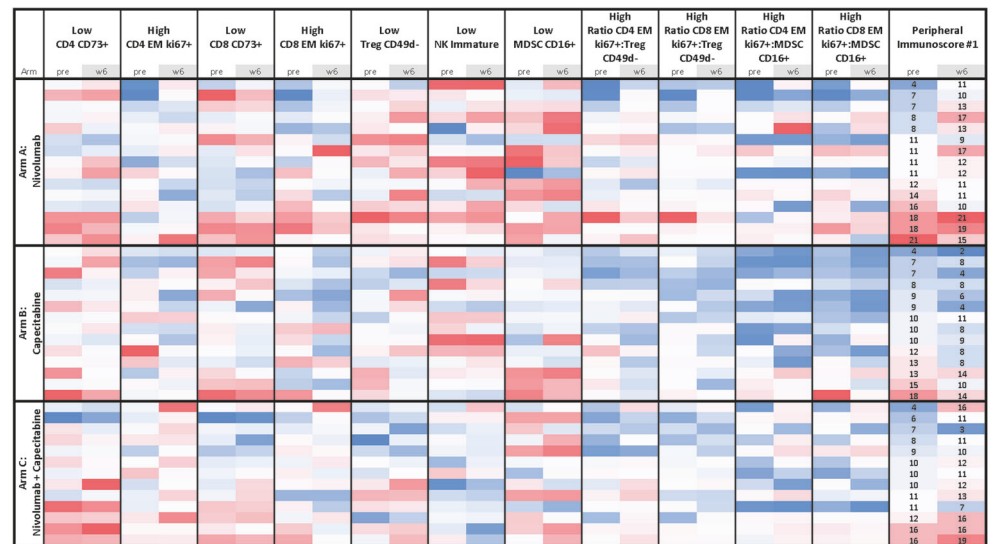

Red: Good immune phenotype reflective of immune activation; Blue: Poor immune phenotype reflective of immune suppression

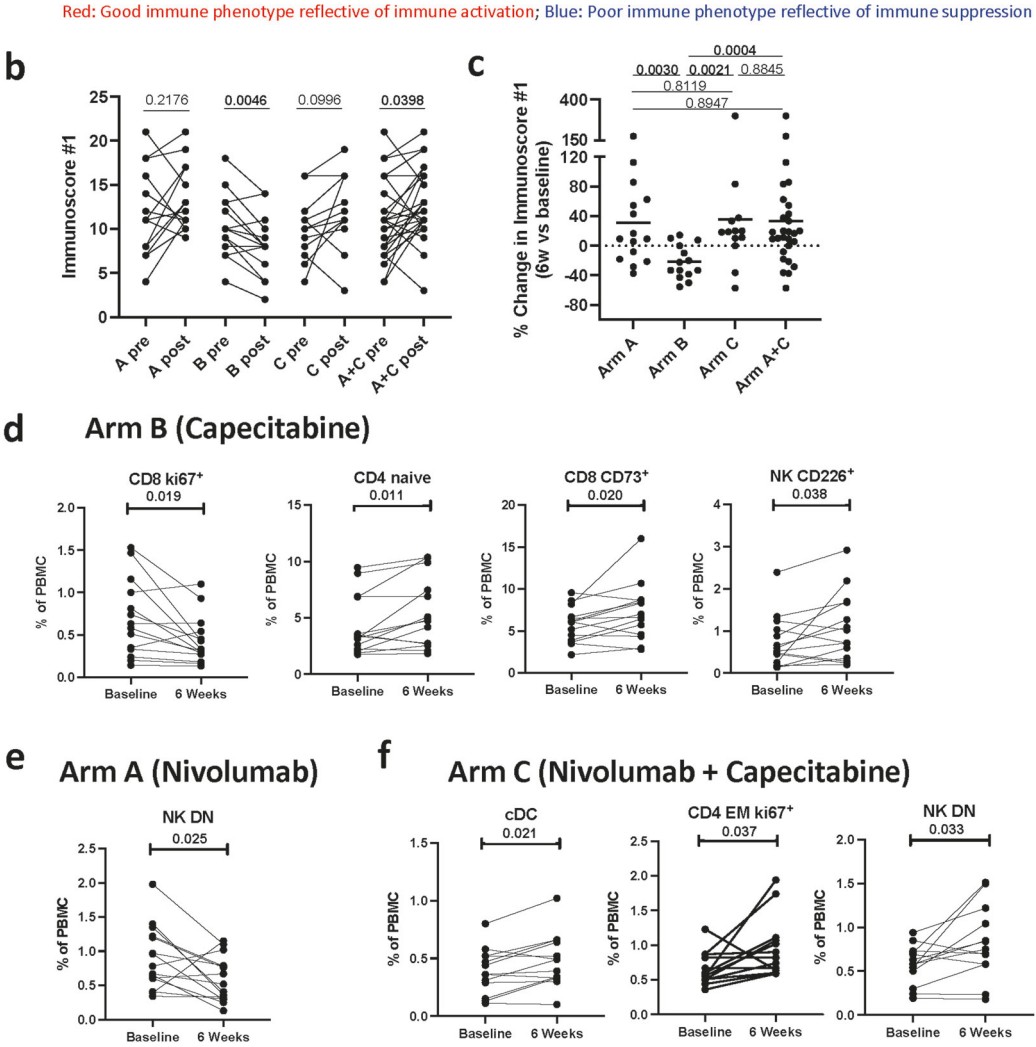

**d** **Arm B (Capecitabine)**

**e** **Arm A (Nivolumab)**

**f** **Arm C (Nivolumab + Capecitabine)**

prior exposure to radiotherapy. For this post-hoc analysis, all patients were combined due to the limited number of patients who had not received prior radiotherapy in each arm (4 in Arm A and Arm B, and 3 in Arm C). Compared to patients who didn't receive radiotherapy, we found that patients who had received prior adjuvant radiotherapy had lower levels of landmark T cells, including total CD4$^+$ T cells ($p = 0.012$),

PD-L1$^+$ CD4$^+$ T cells ($p = 0.016$), ICOS$^+$ CD4$^+$ T cells ($p = 0.004$), central memory (CM) CD4$^+$ T cells ($p = 0.025$), and CM CD8$^+$ T cells ($p = 0.022$) (Supplementary Fig. 4A), higher levels of landmark monocytes, including total monocytes ($p = 0.016$), PD-L1$^+$ monocytes ($p = 0.016$), classical monocytes ($p = 0.041$), PD-L1$^+$ classical monocytes ($p = 0.019$), and intermediate monocytes ($p = 0.016$), and higher levels of pDCs

**Fig. 1 | Changes in peripheral immunoscore #1, and other immune cell subsets after 6 weeks of therapy. a** Heatmap representing the frequency at landmark and 6 weeks of refined classic peripheral blood mononuclear (PBMC) subsets of cell types reflecting known function (Supplementary Table 3) that were used to generate an immunoscore (peripheral immunoscore #1) in patients enrolled in arms A (*n* = 15), B (*n* = 14), C (*n* = 13), and arms A and C combined (*n* = 28). Each row corresponds to one patient. Peripheral immunoscore #1 is the sum of points assigned to each subset based on tertile distribution as previously described[42]. **b** The peripheral immunoscore #1 calculated in A before and after 6 weeks of therapy in each treatment arm and arms A and C combined. **c** Comparison of the percent change after 6 weeks vs. baseline in the peripheral immunoscore in each arm and arms A and C combined. *p* values are shown; *p* values were calculated by a two tailed Wilcoxon signed-rank test in B and a two tailed Mann–Whitney test in C, and no

adjustments were made for multiple comparisons. Additional immune changes in the peripheral immune profile after 6 weeks of treatment in patients treated with (**d**) capecitabine (*n* = 14), **e** nivolumab (*n* = 15), and (**f**) nivolumab plus capecitabine (*n* = 13). For (**d**–**f**), changes in 10 classic PBMC cell types and 148 refined subsets reflective of maturation and function were analyzed with no adjustments made for multiple comparisons. Notable subsets with significant changes at post timepoints vs. landmark are displayed in (**d**–**f**) and include those with *p* < 0.05 (calculated by a two tailed Wilcoxon signed-rank test), difference in medians >0.05, and ≥50% of patients having a >25% change. cDC conventional dendritic cell, DN double negative, EM effector memory, NK natural killer cells, Treg regulatory T cell, MDSC myeloid-derived suppressor cells, PBMC peripheral blood mononuclear cells. Source data are provided as a source data file.

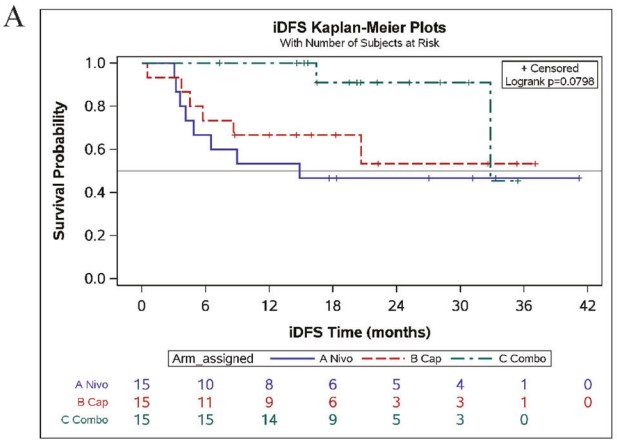

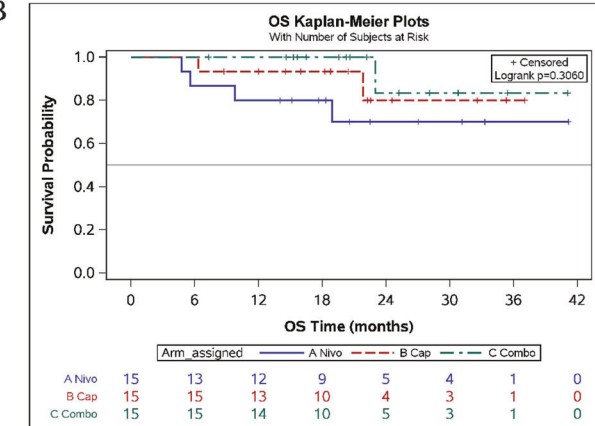

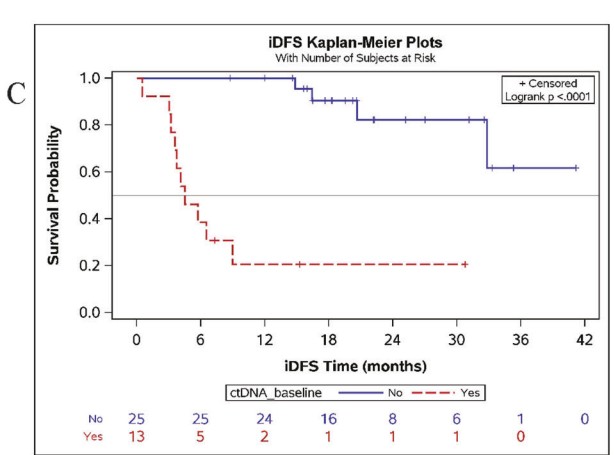

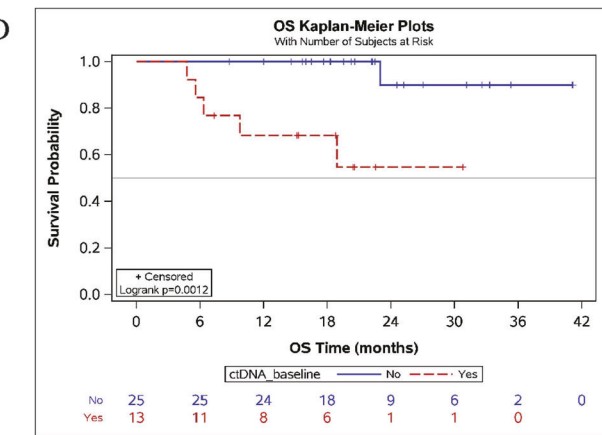

**Fig. 2 | Median invasive disease-free survival (iDFS) and overall survival (OS).** iDFS (**A**) and OS (**B**) stratified by treatment arm. iDFS (**C**) and OS (**D**) stratified by the presence or absence of circulating tumor DNA (ctDNA) at landmark in patients enrolled in Arms A, B, and C combined. Blue line: Arm A (nivolumab); Red line: arm B (capecitabine); Green line: Arm C (combination of nivolumab and capecitabine). Source data are provided as a source data file.

(*p* = 0.006), MDSCs (*p* = 0.020), PD-L1+ MDSCs (*p* = 0.014), ki67+ NK (*p* = 0.047), and NKG2D+ immature NK cells (*p* = 0.026) (Supplementary Fig. 4B).

We also investigated landmark differences in patients with germline *BRCA1* or *BRCA2* (*gBRCA1/2*) mutations compared to patients without known deleterious germline mutations. For this post-hoc analysis, all patients were similarly combined due to the small number of patients with *BRCA1/2* mutations enrolled (2 in Arm A, 1 in Arm B, and 0 in Arm C). While we found a higher percent of CD73+ CD8+ T cells (*p* = 0.005) in patients with *gBRCA1/2* mutations, no other landmark differences were observed (Supplementary Fig. 4C).

**Detection of ctDNA**
Of 45 patients enrolled, 38 patients had sufficient tissue for sequencing with at least 20% tumor present (Supplementary Fig. 5) and underwent

successful WES. Of these, 28 (73.7%) provided residual tissue from breast surgery and 10 (26.3%) from core biopsy. All 38 patients provided at least 1 plasma sample, and 34 provided samples at multiple time points, with a median number of 3 (range 1 to 4) samples per patient and a total of 121 samples. Personalized RaDaR™ assays were designed and applied with 7 to 47 variants included (median 33). Thirteen patients (13/38, 34%) had positive ctDNA at study entry, with variant allele frequencies (VAF) ranging from 0.0012% to 3.6%. Twenty-five percent of ctDNA levels detected at landmark were below 0.01% VAF. The detection of landmark ctDNA differed significantly by clinical stage (*p* = 0.007) and pathological stage (*p* = 0.001) (Table 3). It did not significantly differ by treatment arm, with 46% of ctDNA-evaluable patients in Arm A found to be ctDNA-positive at landmark compared to 33.3% and 23% in Arms B and C, respectively (Supplementary Table 10).

**Table 2 | Most common drug-related adverse events in the safety analysis set**

| Event | Arm A Nivolumab (N = 15) | | Arm B Capecitabine (N = 15) | | Arm C Nivolumab and Capecitabine (N = 15) | |
|---|---|---|---|---|---|---|
| | **All Grades** **Number of patients (%)** | **Grade 3** | **All Grades** | **Grade 3** | **All Grades** | **Grade 3** |
| Endocrine disorders: Hypothyroidism | 3 (20) | 0 | 0 | 0 | 2 (13.3) | 0 |
| Gastrointestinal disorders | | | | | | |
| Abdominal pain | 0 | 0 | 2 (13.3) | 0 | 5 (33.3) | 1 (6.7) |
| Diarrhea | 1 (6.7) | 0 | 7 (46.7) | 0 | 7 (46.7) | 2 (13.3) |
| Nausea | 0 | 0 | 4 (26.7) | 0 | 2 (13.3) | 0 |
| Oral mucositis | 0 | 0 | 3 (20) | 2 (13.3) | 3 (20) | 0 |
| General disorders: fatigue | 6 (40) | 0 | 5 (33.3) | 0 | 4 (26.7) | 2 (13.3) |
| Musculoskeletal and connective tissue disorders: arthralgia | 5 (33.3) | 0 | 1 (6.7) | 0 | 0 | 0 |
| Nervous system disorders: Peripheral sensory neuropathy | 0 | 0 | 3 (20) | 0 | 4 (26.7) | 0 |
| Skin and subcutaneous tissue disorders | | | | | | |
| Palmar-plantar erythrodysesthesia syndrome | 0 | 0 | 7 (46.7) | 0 | 5 (33.4) | 0 |
| Skin and subcutaneous tissue disorders | 0 | 0 | 1 (6.7) | 0 | 4 (26.7) | 0 |

## ctDNA and disease recurrence

At a median follow-up of 20.4 months, among the 38 patients who underwent successful ctDNA testing, 14 experienced a distant recurrence. Ten patients with distant recurrence (71%) were ctDNA-positive at landmark and 11 (79%) were ctDNA-positive at any timepoint. All patients who underwent ctDNA testing at time of recurrence ($n = 5$) were ctDNA-positive (Supplementary Fig. 6).

Patients who were ctDNA-positive at landmark had an inferior median iDFS (4.52 months, 95% CI: 3.21–8.98) compared to patients who were ctDNA-negative (median iDFS: Not Yet Reached; log-rank $p < 0.0001$) (Fig. 2C) (Supplementary Table 11). The median OS was also inferior among patients who were ctDNA-positive at landmark compared to patients who were ctDNA-negative (log-rank $p = 0.0012$) (Fig. 2D).

Among the 13 patients who were ctDNA-positive at landmark, four subsequently cleared ctDNA at 6 weeks. Three of these patients underwent ctDNA testing at 12 weeks. Of those, two remained ctDNA-negative at 12 weeks, whereas one patient became ctDNA-positive 12 weeks after the initiation of therapy and subsequently experienced a distant recurrence. Three out of four patients who became MRD-negative at 6 weeks have not experienced a recurrence to date. The remaining nine patients who had positive ctDNA testing at landmark and did not become ctDNA-negative at 6 weeks all developed distant recurrence (Supplementary Table 11).

Among the 25 patients who were ctDNA-negative at landmark, 24 remained ctDNA-negative at all subsequent timepoints. Only four of these patients (16%) experienced a breast cancer recurrence. One patient had positive ctDNA testing at 6 weeks (VAF of 0.0015%), but became ctDNA-negative at 12 weeks. This patient experienced a distant recurrence.

We performed a post-hoc univariate analysis of iDFS and OS with the following variables: landmark peripheral immunoscore #1 and #2 (dichotomized peripheral immunoscore #1 into "above the baseline median (>10)" vs. "equal or less than the baseline median (<=10)", and dichotomized peripheral immunoscore #2 into "above the baseline median (>11)" vs. "equal or less than the baseline median (<=11)"), treatment arm and ctDNA status. Only landmark ctDNA status was significantly associated with iDFS or OS in all patients combined; landmark peripheral immunoscore #2 was associated with iDFS in certain arms. Therefore, we only performed multivariate analysis for iDFS to further evaluate the effects of landmark ctDNA, landmark peripheral immunoscore #2, and treatment arms. Based on the multivariate Cox proportional hazard model, patients who were ctDNA-positive at landmark had significantly worsen iDFS compared to

patients who were ctDNA-negative: hazard ratio (HR) 50.70 (95% CI: 6.52–393.98, $p < 0.001$). Compared to the patients with a landmark immunoscore #2 equal or below the median (<=11), those with a landmark immunoscore #2 above the median had significantly better iDFS: HR 0.064 (95% CI: 0.009–0.462, $p = 0.006$). When analyzed by treatment arm, patients with landmark immunoscore #2 equal or below the median treated in Arm C experienced significantly improved iDFS compared to those treatment in Arm A (HR 0.027. 95% CI: 0.002–0.35, $p = 0.0058$) but there were no differences between Arm B and Arm A.

## ctDNA and immune profile

We next interrogated whether the immune profile of patients prior to therapy was associated with the presence of ctDNA at landmark. For this post-hoc analysis, all patients were combined due to the limited number of patients with ctDNA in each treatment arm. Individuals who were ctDNA-positive at landmark had higher levels of naïve $CD8^+$ T cells ($p = 0.010$), $CD8^+$ T cells that express CD73 ($p = 0.007$), an immune checkpoint involved in adenosine metabolism, and PD-L1-expressing non-classical monocytes ($p = 0.015$) (Fig. 5A). Among the group of patients who were ctDNA-negative at landmark and underwent immune profiling ($n = 24$), we evaluated whether there were landmark immunologic differences in those patients who recurred ($n = 4$) and did not recur ($n = 20$). We found higher levels of multiple refined Treg subsets, including those expressing ki67 ($p = 0.018$), HLA-DR ($p = 0.007$), and ICOS ($p = 0.045$) at landmark in ctDNA-negative patients who experienced a recurrence (Fig. 5B).

Next, among the group of ctDNA-positive patients at landmark who underwent immune profiling ($n = 11$), we investigated whether there were immunologic differences at the landmark timepoint in those patients who recurred ($n = 9$) and did not recur ($n = 2$). Here, we found lower frequencies of total NK cells ($p = 0.036$), mature ($CD56^{dim}$ $CD16^+$) NK cells ($p = 0.036$), NK cells that express the activating receptor NKp46 ($p = 0.036$), and PD-1$^+$ NKT cells in patients who recurred compared to those who did not (Fig. 5c). Notably, the two patients with detectable ctDNA at landmark who did not develop a recurrence cleared their ctDNA during therapy.

## Discussion

This randomized phase II study was designed to evaluate the role of adjuvant nivolumab, capecitabine or the combination for the treatment of patients with early-stage TNBC with residual invasive disease after the completion of neoadjuvant chemotherapy. The study met the pre-specified primary endpoint, with patients treated with

## a    Arm A (Nivolumab) + B (Capecitabine) + C (Nivolumab + Capecitabine)

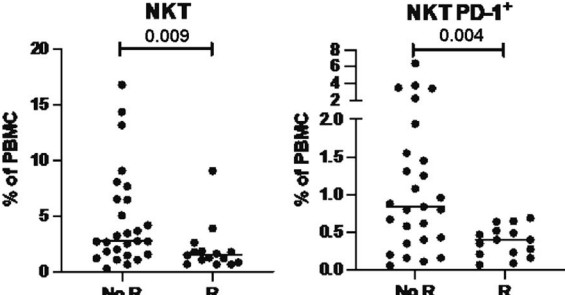

## b    Arm A (Nivolumab)

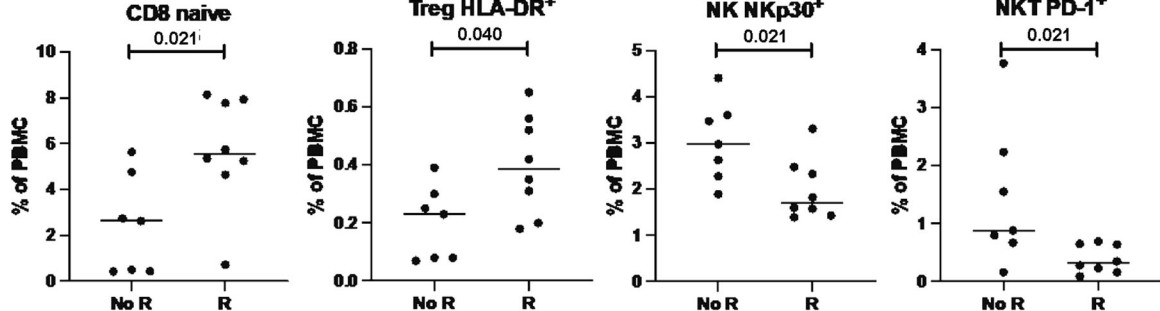

## c    Arm B (Capecitabine)

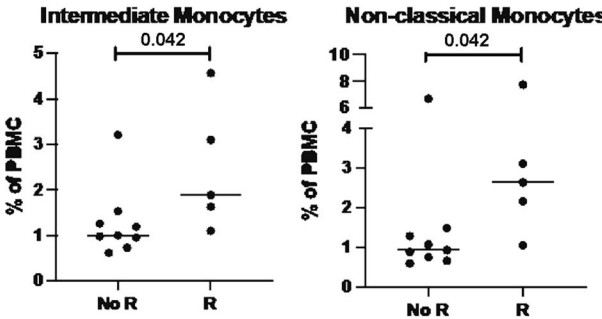

## d    Arm C (Nivolumab + Capecitabine)

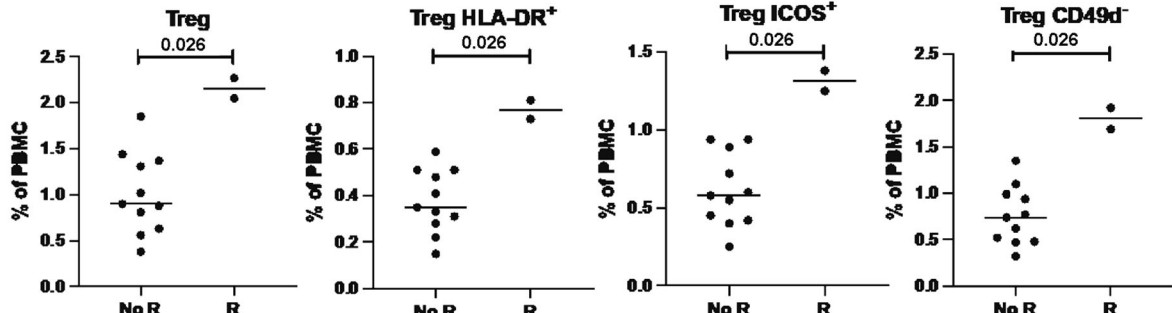

**Fig. 3 | The peripheral immune profile at landmark associates with the development of recurrence after therapy.** The peripheral immune profile at landmark was compared between patients who developed a recurrence (R) and those that did not (no R). Frequency of immune subsets at landmark that associate with recurrence in patients treated with (**a**) nivolumab, capecitabine, or nivolumab + capecitabine ($n = 27$ with no R, $n = 15$ with R), **b** nivolumab ($n = 7$ with no R, $n = 8$ with R), **c** capecitabine ($n = 9$ with no R, $n = 5$ with R), and (**d**) nivolumab plus capecitabine ($n = 11$ with no R, $n = 2$ with R). Differences in 10 classic peripheral blood mononuclear (PBMC) cell types and 148 refined subsets reflective of maturation and function were analyzed. Notable subsets with significant differences are displayed and include those with $p < 0.05$ (calculated by a two tailed Mann–Whitney test), and difference in medians >0.05 of PBMCs. No adjustments were performed for multiple comparisons. NK/NKT natural killer T cells, PD-1 programmed cell death protein 1, Treg regulatory T cell, HLA-DR Human Leukocyte Antigen – DR isotype, ICOS inducible T cell co-stimulator, PBMC peripheral blood mononuclear cells. Source data are provided as a Source Data File.

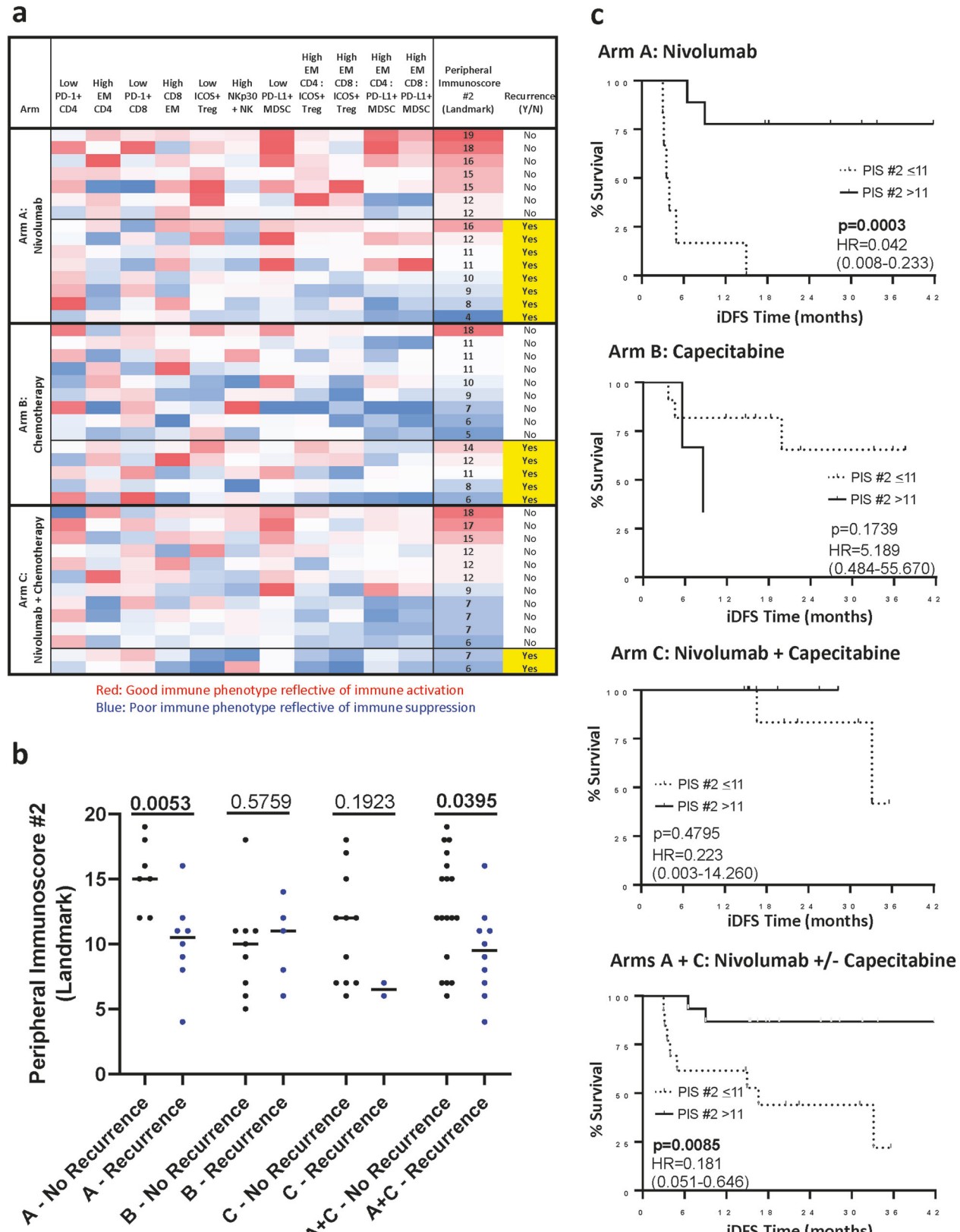

Red: Good immune phenotype reflective of immune activation
Blue: Poor immune phenotype reflective of immune suppression

immunotherapy containing regimens (arms A and C) experiencing an increase at week 6 versus baseline in a peripheral immunoscore (immunoscore #1) compared to patients treated with chemotherapy alone (Arm B). The combination regimen was associated with a numerical improvement in median iDFS and OS compared to nivolumab or capecitabine monotherapy, although the difference did not reach statistical significance and this study was not powered for survival endpoints.

On retrospective analysis, the three study arms had an unbalanced distribution of ctDNA-positive patients, potentially contributing to the differences noted in clinical outcomes. As previously reported, checkpoint inhibitor given as monotherapy was better tolerated when

**Fig. 4 | Peripheral immunoscore #2 at landmark associates with disease recurrence in patients receiving nivolumab or nivolumab +/− chemotherapy. a** Heatmap representing the frequency at landmark of refined peripheral blood mononuclear cell (PBMC) subsets of cell types reflecting known function (Supplementary Table 9) that were used to generate an immunoscore (peripheral immunoscore #2) in patients enrolled in arms A (n = 15), B (n = 14), C (n = 13), and arms A and C combined (n = 28). Each row corresponds to one patient. The peripheral immunoscore #2 is the sum of points assigned to each subset based on tertile distribution as previously described[42]. **b** Association between the peripheral immunoscore #2 calculated in A with disease recurrence following therapy in each arm and arms A and C combined. Peripheral immunoscore #2 was compared in patients with no disease recurrence (no R) vs patients with disease recurrence (R) in

Arm A (n = 7 no R, n = 8 R), Arm B (n = 9 no R, n = 5 R), Arm C (n = 11 no R, n = 2 R), and Arms A + C combined (n = 18 no R, n = 10 R). Medians with p values are shown; p values were calculated by a two tailed Mann−Whitney test. **c** Association between the peripheral immunoscore #2 calculated in A and iDFS in arms A (n = 15), B (n = 14), C (n = 13), and arms A and C combined (n = 28), were analyzed using a Log-rank (Mantel-Cox) test. Hazard ratio and 95% confidence interval, calculated by the Mantel−Haenszel method, are indicated. Solid line: patients with peripheral immunoscore #2 (PIS #2) >the median; dashed line: patients with PIS #2 ≤ the median. PD-1 programmed cell death protein 1, EM effector memory, NK natural killer cells, NKp 30 natural killer cells activating receptor 30, ICOS inducible T cell co-stimulator, Treg regulatory T cell, MDSC myeloid-derived suppressor cells. Source data are provided as a Source Data File.

**Table 3 | Landmark circulating tumor DNA (ctDNA) by treatment arm and disease stage for patients with available landmark ctDNA results (N = 38)**

| Variable | | Overall (N = 45) | Landmark ctDNA | | p-value |
|---|---|---|---|---|---|
| | | | Yes (N = 13) | No (N = 25) | |
| Arm Assigned | A: Nivolumab | 15 (33.3%) | 6 (46.2%) | 7 (28.0%) | 0.482 |
| | B: Capecitabine | 15 (33.3%) | 4 (30.8%) | 8 (32.0%) | |
| | C: Combination | 15 (33.3%) | 3 (23.1%) | 10 (40.0%) | |
| Clinical Staging | IA | 2 (4.4%) | 1 (7.7%) | 1 (4.0%) | 0.007 |
| | IB | 1 (2.2%) | 0 (0%) | 1 (4.0%) | |
| | IIA | 17 (37.8%) | 1 (7.7%) | 13 (52.0%) | |
| | IIB | 7 (15.6%) | 5 (38.5%) | 2 (8.0%) | |
| | IIIA | 13 (28.9%) | 6 (46.2%) | 5 (20.0%) | |
| | IIIB | 5 (11.1%) | 0 (0%) | 3 (12.0%) | |
| Pathological Staging | IA | 10 (22.2%) | 0 (0%) | 10 (40.0%) | 0.001 |
| | IB | 3 (6.7%) | 0 (0%) | 2 (8.0%) | |
| | IIA | 14 (31.1%) | 3 (23.1%) | 7 (28.0%) | |
| | IIB | 6 (13.3%) | 3 (23.1%) | 3 (12.0%) | |
| | IIIA | 6 (13.3%) | 2 (15.4%) | 3 (12.0%) | |
| | IIIC | 6 (13.3%) | 5 (38.5%) | 0 (0%) | |

All p-values were calculated from the two-sided Fisher's exact test without adjustment.

compared to single agent chemotherapy[74]. The dose intensity of capecitabine was similar in both capecitabine-containing arms, suggesting that the combination with immunotherapy did not impair the administration of capecitabine. Although no new safety signals were identified in the combination arm, the incidence of drug-related adverse events, including grade 3, was higher in the combination arm, suggesting there may be some degree of synergistic toxicity between nivolumab and capecitabine. A phase II single arm study of capecitabine and pembrolizumab in patients with HER2-negative advanced breast cancer reported that the combination was well tolerated, and most observed adverse events were low grade and consistent with what would be expected with capecitabine monotherapy[75]. However our results are aligned with what was observed in the CHECKMATE 649 trial, in which patients with treatment-naïve, HER2-negative, unresectable gastric, gastro-esophageal, or esophageal adenocarcinoma were randomized to receive nivolumab plus chemotherapy (capecitabine and oxaliplatin every 3 weeks or leucovorin, fluorouracil, and oxaliplatin every 2 weeks) or chemotherapy alone. The rate of grade 3–4 treatment-related adverse events was 59% among patients who received nivolumab plus chemotherapy compared to 44% among patients who received chemotherapy alone[76]. In the present study, there were no grade 4 or grade 5 toxicities observed and importantly, there was no increase in irAEs in the combination arm.

Numerous prior studies have demonstrated that the analysis of circulating immune cells can provide potentially prognostic information about therapeutic effectiveness[36–41,43,77,78]. For example, in a cohort

of patients undergoing first-line systemic therapy for advanced breast cancer, those with clinical benefit had an increase in peripheral activated T cells and decreased Tregs, MDSCs, and PD-1-expressing T cells[79]. In addition, several studies in patients with breast cancer have attempted to capture changes in the peripheral immune system during neoadjuvant chemotherapy[80–82]. In the present study, we evaluated whether a peripheral immunoscore (immunoscore #1) reflective of enhanced immune function, that was based on the frequency and ratios of immune subsets at landmark and 6 weeks with well-known biologic function[55–64], was differently changed after therapy in each arm. We also assessed the effects of each treatment on classic and refined PBMC subsets. Clear differences emerged in each treatment arm in terms of effects on immune cells. Among the 158 PBMC subsets evaluated, patients who received nivolumab only had a decrease in circulating double negative (CD56$^{dim}$CD16$^-$) NK cells. This refined NK subset has been shown to be cytotoxic against tumor cells in vitro[83]; however, more recently this subset has also been described as a non-cytolytic and exhausted NK subset enriched in TNBC patients with residual disease after surgery[65]. In general, more extensive changes in peripheral subsets were seen in patients who received capecitabine, either alone or in combination with nivolumab. With capecitabine monotherapy, we observed notable reductions in proliferative CD8$^+$ T cells, an immune inhibitory effect, and increases in both CD73$^+$CD8$^+$ T cells and CD226$^+$ NK cells, which could have variable implications for anti-cancer immunity. In pre-clinical studies, CD73 has been shown to restrict the cytotoxic anti-tumor activity of CD8$^+$ T cells[84], while

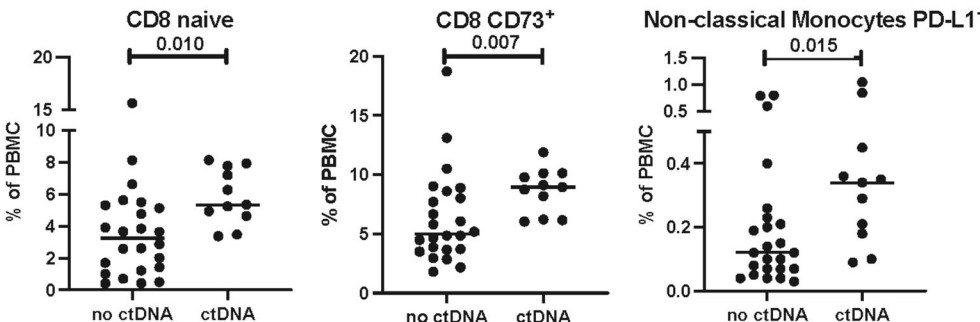

## a  Absence vs Presence of ctDNA at baseline

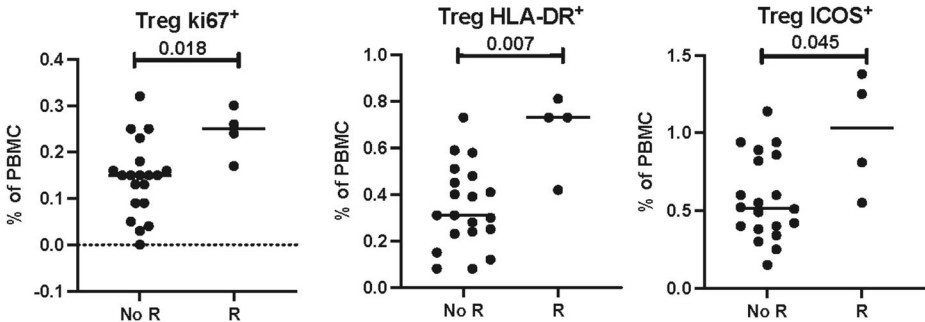

## b  Recurrence in patients *without* ctDNA at baseline

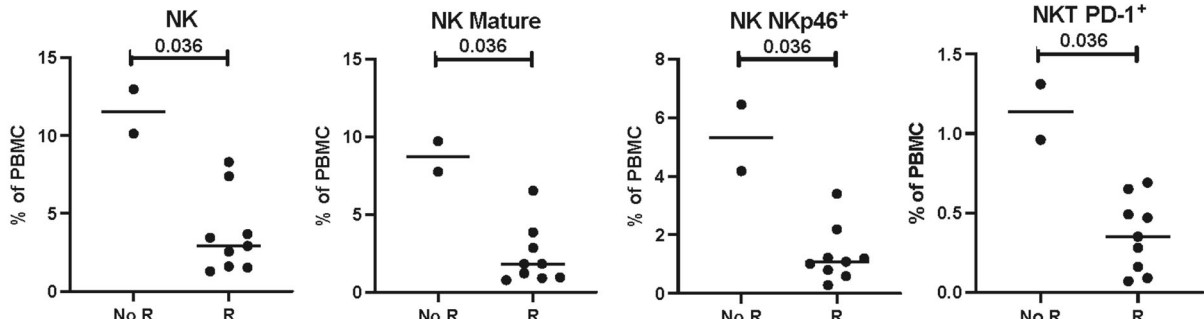

## c  Recurrence in patients *with* ctDNA at baseline

**Fig. 5 | Association of the peripheral immune profile at landmark in patients from Arms A, B and C combined with the presence of ctDNA at landmark and recurrence.** The peripheral immune profile was compared at landmark in all arms combined between patients with presence and absence of landmark ctDNA. Frequency of PBMC subsets at landmark that differed between (**a**) patients with ($n = 11$) and without ($n = 24$) ctDNA at landmark, **b** patients without ctDNA at landmark who recurred (R, $n = 4$) vs. did not recur (no R, $n = 20$) after therapy, and (**c**) patients with ctDNA at landmark who recurred (R, $n = 9$) and did not recur (no R, $n = 2$) following therapy. Differences were analyzed in 10 classic PBMC cell types and 148 refined PBMC subsets reflective of maturation and function. Notable subsets with significant differences are displayed and include those with $p < 0.05$ (calculated by a two tailed Mann–Whitney test), and a difference in medians >0.05 of PBMCs. No adjustments were made for multiple comparisons. ctDNA circulating tumor DNA, PBMC peripheral blood mononuclear cells, PD-L 1 programmed death-ligand 1, Treg regulatory T cell, HLA-DR Human Leukocyte Antigen – DR isotype, ICOS inducible T cell co-stimulator, NK natural killer cells, NKp46 natural cytotoxicity triggering receptor 1. Source data are provided as a Source Data File.

CD226, an adhesion molecule and activating receptor has been shown to be important for NK cell anti-tumor activity in vitro[85] and in patients with cancer[86]. Patients receiving nivolumab plus capecitabine had increases in double negative NK cells, cDCs which are involved in antigen presentation, and proliferative effector memory CD4$^+$ T cells subsets. These findings suggest that some of the immune inhibitory actions of capecitabine may have been alleviated by the addition of nivolumab to capecitabine. In a prior study, neoadjuvant chemotherapy caused a transient increase in NK cells, NKT cells, and non-classical monocytes in breast cancer patients[80]. Our results further support that chemotherapy in combination with an ICI causes extensive changes in the peripheral immune system that may enhance anti-tumor activity.

We also showed that the immune profile of patients prior to therapy was associated with the development of recurrence. These studies were based on the idea that the immune cell profile of a given patient, as detected in peripheral blood, may affect their response to immune-mediated therapy. Differences in the immune profile could be influenced by a number of factors, including the type(s) and line(s) of prior therapy, tumor type and stage, tumor size, the microbiome, stress, and genetic factors. In immune analyses combining all treatment arms, individuals who developed recurrence had lower landmark

levels of NKT cells compared to patients who did not recur. NKT cells, depending on the tumor model system evaluated, have been shown to contribute to both immunostimulatory and immunosuppressive anti-tumor responses[87]. Of particular interest, higher landmark levels of Tregs were observed in patients on nivolumab monotherapy or nivolumab plus capecitabine who experienced metastatic recurrence, relative to those who remained without evidence of disease progression. This pattern was also observed in the subset of patients who were ctDNA-negative at landmark and went on to experience metastatic recurrence. Tregs are well known to suppress the anti-tumor activity of CD8$^+$ T cells, CD4$^+$ T cells, and other cytotoxic immune cells[88]. The prognostic significance of Treg enrichment in the tumor microenvironment is controversial, particularly in TNBC[89,90], and it is not known how the presence of Tregs in the peripheral blood correlates with the tumor microenvironment. Further work will be necessary to unravel the prognostic significance of decreased NKT cells and increased circulating Tregs in early-stage TNBC patients with residual invasive disease. We also showed in an exploratory analysis, that the calculation of a second peripheral immunoscore (immunoscore #2), based on the pre-therapy frequencies of specific refined PBMC subsets, each of which has been shown in the literature to have immune enhancing or immune suppressing functions[56,66–73], may help to identify those breast cancer patients in larger randomized studies employing these regimens. These results support the rationale for the interrogation of prospective immune profiling to identify patients who are at high risk of recurrence after neoadjuvant and/or adjuvant therapy. It should be noted and is logical that the immune cell components comprising peripheral immunoscore #1 and peripheral immunoscore #2 differ, given that immunoscore #1 evaluates the change of immune components as a result of different therapies, while immunoscore #2 analyzes the general immune status of patients only prior to therapy.

As mentioned above, prior therapy has the potential to induce systemic changes to the immune system; therefore, in this population where 76% of patients had received prior radiotherapy, we investigated whether differences existed in the landmark immune profile of patients who received versus did not receive prior radiotherapy. We found that patients who received prior adjuvant radiotherapy had decreased levels of total CD4$^+$ and refined CD4$^+$ T cell subsets and CM CD8$^+$ T cells compared to those who didn't receive radiotherapy. Prior studies also report reduction in CD4$^+$ T cells following RT in patients with various solid tumors[91]. Interestingly, we also observed increases in peripheral immune subsets of monocytes and MDSCs in patients who received prior radiotherapy. Monocytes have been found to be more resistant to radiotherapy than lymphocytes[92,93], and expansions in MDSCs have also been noted following radiotherapy[94].

For patients with early-stage breast cancer, distant recurrence likely arises from residual cancer cells remaining after curative intent therapy that are not detected via standard imaging, laboratory tests, or clinical assessment[95]. Several studies in patients with early-stage breast cancer have shown that the detection of ctDNA can portend the diagnosis of distant recurrence by 1 year or more[24–27,29,30,32,96]. In agreement with these studies, we found that ctDNA-positive patients at landmark who did not clear ctDNA during treatment all experienced distant recurrence in a relatively short period of follow-up time. Moreover, ctDNA positivity at landmark was associated with worse survival outcomes. These results support the continued development of prospective ctDNA monitoring to identify ctDNA-positive patients who are at high risk of recurrence after neoadjuvant and/or adjuvant therapy.

One of the major strengths of the current study is the evaluation of peripheral immune cell subsets and ctDNA before the initiation of therapy, at multiple timepoints during therapy, and at recurrence. Both the peripheral immune composition of patients and ctDNA positivity at landmark were associated with the development of

recurrence. Capturing changes in the immune system during treatment can help to identify evolving mechanisms of tumor immune escape. It also adds to a growing body of evidence that ctDNA positivity is associated with increased risk of recurrence.

This study was limited by the small sample size, with 15 patients on each treatment arm for potential evaluation of ctDNA and peripheral immune cell subsets, leading to unbalanced distribution of ctDNA positivity and peripheral immune subsets in each arm. Moreover, the study was not statistically powered to assess differences in survival outcomes between treatment arms or different landmark immune subsets. Thus, findings from this study are exploratory and should be interpreted as hypothesis-generating. In addition, unlike in KEYNOTE-522[15], none of these patients received an ICI in the neoadjuvant setting, given that pre-operative chemo-immunotherapy was not standard of care when this trial was enrolling patients[15]. It is unknown how exposure to an ICI in the neoadjuvant setting might influence the results of landmark (pre-adjuvant) peripheral immune subsets as well as prevalence of landmark ctDNA positivity. Given the findings of the KEYNOTE-522 trial[15], as well as the positive results of SWOG S1801 trial[97], which compared neoadjuvant pembrolizumab followed by adjuvant pembrolizumab to adjuvant pembrolizumab alone among patients with stage IIIB to IVC melanoma, it is possible that most of the benefit achieved with checkpoint inhibitors may be obtained in the neoadjuvant setting.

In summary, the translational findings from the OXEL study support the continued development of prospective immune profiling and ctDNA monitoring as a means of identifying early-stage TNBC patients who are at high risk of recurrence following neoadjuvant or adjuvant therapy. Further study of peripheral immune cell subsets at landmark and during therapy may help to identify molecular targets to improve the efficacy of adjuvant therapy for patients at higher risk of recurrence. An algorithm incorporating landmark ctDNA analysis and peripheral immune cell subsets may better identify patients with TNBC at higher risk of recurrence. Future trials aiming to optimize adjuvant therapy with chemotherapy and/or immunotherapy in residual TNBC should consider incorporating ctDNA as a selection marker of patients at higher risk of recurrence or assure that treatment arms are balanced for ctDNA-positivity status.

## Methods

### Study design and patient population
OXEL (https://classic.clinicaltrials.gov/ct2/show/NCT03487666, pre-registered on April 4, 2018) was a phase II, open-label, multi-institutional trial that enrolled patients with early-stage TNBC defined as ER ≤ 5%, PR ≤ 5%, and HER2-negative with residual invasive disease of at least 1.0 cm in the breast and/or positive lymph node(s) (at least ypN1) after completion of neoadjuvant chemotherapy between August 2018 and June 2021. Anthracycline, taxane, and/or carboplatin-containing neoadjuvant chemotherapy regimens were allowed. Preoperative immunotherapy was not allowed. Participants must have completed definitive resection of primary tumor and had no evidence of metastatic disease at the time of study entry. Staging scans prior to study entry were not required. This study complied with all relevant ethical regulations, including the Declaration of Helsinki, and was approved by the MedStar Georgetown University Hospital Institutional Review Board. Patients were enrolled at MedStar Georgetown University Hospital, MedStar Washington Hospital Center, University of Chicago, University of Alabama Birmingham, and Hackensack University Medical Center. Men and women were both eligible for this trial, but only women were enrolled. All patients provided informed consent.

### Treatment and follow-up
Participants were randomized 1:1:1 to receive nivolumab (Arm A), capecitabine (Arm B), or a combination of nivolumab and capecitabine

(Arm C). Nivolumab 360 mg intravenous (i.v.) was administered once every 3 weeks for six cycles. Capecitabine 1250 mg/m² oral was administered twice daily on days 1–14 of each 3-week cycle for six cycles. Nivolumab was provided by Bristol-Myers Squibb and capecitabine was commercially obtained. Participants were followed for recurrence by their physicians using routine follow-up visits and breast imaging standard of care (Supplementary Fig. 7). ctDNA results were analyzed retrospectively.

## Study endpoints

The primary endpoint was to assess the effects of nivolumab, capecitabine, or the combination on the peripheral immune profile. We hypothesized that among patients with TNBC and residual disease at the time of surgery, the change of a Peripheral ImmunoScore (PIS) from landmark to week 6 will be higher among those who receive post-surgery immunotherapy (Arm A and C), compared to those who receive post-surgery chemotherapy alone (Arm B). Detection of ctDNA at landmark, at 6 weeks, and at 12 weeks was a secondary endpoint. Additional secondary endpoints included incidence of toxicity using the NCI CTCAE v.4.0, OS and iDFS at 2 years, and association between ctDNA and peripheral immune profile with recurrence and survival. iDFS was defined as the time from date of randomization to the date of first invasive disease recurrence, second invasive primary cancer (breast or not), or death from any cause. OS was defined as the time from date of randomization to death from any cause. A second peripheral immunoscore based on specific refined immune cell subsets at landmark (defined as immunoscore #2) was evaluated as an unplanned and exploratory endpoint.

## Research biospecimens

Primary archival tumor tissue was collected from diagnosis and from time of definitive breast surgery. Serial blood samples (30 mL in Streck tubes) were collected at landmark (before the initiation of therapy), 6 weeks, 12 weeks, and at time of recurrence (if applicable). PBMCs were isolated and cryopreserved.

## Peripheral immune cells

Cryopreserved PBMCs collected from patients at landmark, 6 weeks, and 12 weeks were examined by multicolor flow cytometry using 30 markers in four panels[43] to identify 158 peripheral immune cell subsets with known biologic function (Supplementary Table 2) following methods previously described[98,99], and using the gating strategy shown (Supplementary Fig. 8). Antibodies used to detect 10 parental cell types (CD4⁺ and CD8⁺ T cells, Tregs, NK cells, NKT cells, cDCs, plasmacytoid dendritic cells (pDCs), B cells, MDSCs, and monocytes), and 148 refined subsets related to the maturation/function of the parental cell types by flow cytometry are indicated (Supplementary Table 12). PD-1-expressing subsets were not included in the analyses after treatment with nivolumab as the anti PD-1 clone utilized in the current study (EH12.2H7) recognizes an epitope of PD-1 that is shared with nivolumab[57,100,101]. Flow cytometry files were acquired on an LSRFortessa (BD Biosciences, Franklin Lakes, NJ) equipped with five lasers and analyzed using FlowJo v.9.9.6 (FlowJo LLC, Ashland, OR) for Macintosh, with nonviable cells excluded and negative gates based on fluorescence-minus-one controls. The frequency of all immune subsets was calculated as a percentage of total PBMCs to eliminate any bias that might occur in the smaller populations with fluctuations in parental leukocyte subpopulations. The change in the immune profile was determined by evaluating distinct immune subsets for statistical changes after therapy. Peripheral immune subsets with changes following therapy were defined as those with a $p < 0.05$, ≥50% of patients having a >25% change, and difference in medians of pre- vs post-therapy >0.05% of PBMCs. Immune subsets with median values comprising <0.01% of total PBMCs were excluded from analyses in an effort to focus on potentially biologically relevant immune subsets.

Two peripheral immunoscores were developed using methods previously described[42] based on tertile distribution of frequencies and ratios of peripheral immune cell subsets in patients prior to therapy (Immunoscore #1 and #2). Immune subsets were calculated as a % of PBMC and sorted by frequency. Points were assigned to each subset in a given patient based on tertile distribution. For subsets with an expected positive effect on anti-tumor immunity zero (0) points were assigned to the low bin, one (1) point for the middle bin, and two (2) points if in the high bin. For subsets with an expected negative effect on anti-tumor immunity zero (0) points were assigned to the high bin, one (1) point for the middle bin, and two (2) points if in the low bin. The peripheral immunoscore for a given patient was the sum of points assigned to the individual PBMC subsets that were included within the immunoscore.

Peripheral immunoscore #1 was evaluated at landmark, 6, and 12 weeks for changes with therapy (Fig. 1, Supplementary Fig. 2). It consisted of the % of ki67⁺ Effector Memory (EM) CD4⁺ T cells and ki67⁺ EM CD8⁺ T cells, which have been shown to positively associate with immunity[56,57], the % of CD73⁺ CD4⁺ T cells, CD73⁺ CD8⁺ T cells, immature (CD56^br, CD16⁻) NK cells, CD49d⁻ Tregs and CD16⁺ MDSCs, all of which have been shown to negatively associate with immunity[55,58–64], and the ratio of ki67⁺ EM CD4⁺ T cells: CD49d⁻ Tregs, ki67⁺ EM CD8⁺ T cells: CD49d⁻ Tregs, ki67⁺ EM CD4⁺ T cells: CD16⁺ MDSC, and ki67⁺ EM CD8⁺ T cells: CD16⁺ MDSC, which are expected to positively associate with immunity (Supplementary Table 3). Peripheral immunoscore #2 was evaluated only at landmark in association with disease recurrence (Fig. 4). It consisted of the % of EM CD4⁺ T cells, EM CD8⁺ T cells, and NKp30⁺ NK cells, all of which have been shown to positively associate with immunity[56,67,73], the % of PD-1⁺ CD4⁺ T cells, PD-1⁺ CD8⁺ T cells, ICOS⁺ Tregs and PD-L1⁺ MDSCs, all of which have been shown to negatively associate with immunity[66,68–72], and the ratio of EM CD4⁺ T cells: ICOS⁺ Tregs, EM CD8⁺ T cells: ICOS⁺ Tregs, EM CD4⁺ T cells: PD-L1⁺ MDSC, and EM CD8⁺ T cells: PD-L1⁺ MDSC, which are expected to positively associate with immunity (Supplementary Table 9).

## ctDNA detection

Archival tumor tissue was obtained preferentially from definitive breast cancer surgery, or from initial diagnostic biopsy if tissue was insufficient. Formalin-fixed paraffin-embedded (FFPE) tissue block or 10–20 unstained slides and hematoxylin and eosin stain (H&E) slide from each patient were sent to Inivata, Inc. (Durham, NC), where DNA was extracted and WES was performed as previously described[102,103]. The unique somatic mutation profile of each tumor was used to design a personalized RaDaR™ assay to detect ctDNA in plasma samples from each patient[102–104]. Blood samples were sent to Inivata at time of collection, spun, and then stored at −80 °C as plasma and buffy coat. DNA was extracted and RaDaR™ assays were applied retrospectively in a research setting. Given that this testing was performed retrospectively and designated for research purposes only, patients and their care teams were not informed of the results.

## Safety

Adverse events were coded and graded according to the National Cancer Institute Common Terminology Criteria for Adverse Events (CTCAE) version 4.0.

## Statistical analyses

The sample size of 45 patients with 15 patients per arm had an 85% power to detect an effect size of 1 (the difference of the change in peripheral immunoscore from landmark to week 6 between two arms divided by the standard deviation) at 5% significance level. Stratified randomization was used to assign patients into the three arms (nivolumab, capecitabine and nivolumab/capecitabine combo). Within each stratum, blocked randomization with randomly selected block sizes was used. The stratified randomization procedure was carried out by

the biostatistician(s) at the LCCC Biostatistics and Bioinformatics Shared Resource. Statistical analyses were performed using SAS Software Version 9.4 (SAS Inc, Cary, NC), RStudio (Version 1.4.1106) and GraphPad Prism (GraphPad Software, La Jolla, CA). Changes in immune parameters between two timepoints were assessed for statistical significance using a Wilcoxon signed-rank test. Immune parameters were compared between groups of patients who did or did not recur following therapy, or who did or did not have detectable levels of ctDNA using Chi-square test, Fisher's exact test, $t$-test, or Mann–Whitney test when appropriate. iDFS and OS were analyzed by the Kaplan–Meier method and the log-rank test was used to compare iDFS and OS according to landmark ctDNA results. The relative dose (RD) ([actual total dose/intended total dose] * 100) and the relative dose intensity (RDI) ([actual overall dose intensity/intended overall dose intensity] * 100) of capecitabine were analyzed post hoc. All $p$-values were two-tailed and reported without adjustment for multiple comparisons in this hypothesis-generating study; $p < 0.05$ were considered statistically significant.

### Reporting summary

Further information on research design is available in the Nature Portfolio Reporting Summary linked to this article.

## Data availability

Summarized clinical data, peripheral immune subset data and the clinical trial protocol are provided as Supplementary Information. Additional de-identified participant data are available for academic purposes on request from the corresponding author, Dr. Filipa Lynce (filipa_lynce@dfci.harvard.edu). According to Georgetown IRB authorization based on patients' consent to share genomic data, 29 patients out of 38 with genomic data available consented to have their data deposited. The WES data of 29/38 patients have been deposited at the European Genome-phenome Archive (EGA), which is hosted by the EBI and the CRG, under accession number EGAS50000000222. Controlled access is required to ensure that data use is not for profit or commercial purposes. Data are available by submitting a data access request via the EGA portal (see https://ega-archive.org/access/request-data/how-to-request-data/ for detailed guidance). The remaining data are available within the article, Supplementary Information or Source Data file. Source data are provided with this paper.

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

## Acknowledgements

This study was supported by research grant P30CA051008-25 from the National Cancer Institute (NCI), the Schulbe Family/Nina Hyde Center for Breast Cancer Research, a Partners in Research grant from Georgetown University, the Intramural Research Program Center for Cancer

Research, NCI, the Benderson Family Funds (FL), the Forget Me Not Fund (HAP), and Inivata. Bristol-Myers Squibb provided nivolumab. The funders did not play a role in the study design, data collection and analysis, or manuscript writing. The sponsor investigator is Dr. Filipa Lynce. The authors acknowledge Timothy K. Erick and Valerie Hope Goldstein, full-time employees of Dana-Farber Cancer Institute, for editorial and submission assistance in the preparation of this manuscript.

## Author contributions

F.L., C.M., and R.N.D. contributed equally as co-first authors. Conception and Design: F.L., X.G., G.J., H.W., C.G., S.M.S., P.P., C.I., J.S., R.N.D., and M.G.M. Provision of study materials or patients: F.L., CM., G.J., J.Z., C.G., R.Na, D.G., E.M.S.R., N.D., J.C., A.C., S.T., R.Nu, R.K., K.K., P.P., C.I., and J.S. Collection and assembly of data: F.L., X.G., H.W., R.N.D., N.J.T., C.J., and I.S. Data analysis and interpretation: F.L., R.N.D., N.J.T., C.J., J.S., F.L., X.G., G.J., H.W., P.T., S.M.T., and H.A.P. Manuscript writing and final approval of manuscript: All authors. Accountable for all aspects of work: All authors.

## Competing interests

F.L. reports consulting/advisory role for AstraZeneca, Pfizer, Merck and Daiichi Sankyo; and institutional research funding from Eisai, AstraZeneca, CytomX and Gilead Sciences. M.G.M. reports consulting/advisory roles for GE and Seagen. G.J. reports being a shareholder and employee of NeoGenomics. C.G. reports serving on advisory boards for AstraZeneca, Daiichi Sankyo and Lilly Oncology. R.Na reports serving on advisory boards for AstraZeneca, BeyondSpring, Fujifilm, GE, Gilead, Infinity, iTeos, Merck, OBI, Oncosec, Sanofi and Seagen; and reports research funding from Arvinas, AstraZeneca, Celgene, Corcept Therapeutics, Genentech/Roche, Gilead/Immunomedics, Merck, OBI Pharma, Onco-Sec, Pfizer, Relay, Seattle Genetics, Sun Pharma and Taiho. E.S.-R. reports serving as a consultant for Lilly, Mylan, Novartis, Immunomedics, AstraZeneca, Seagen, and Merck; and institutional research funding from Susan G. Komen, V Foundation, Breast Cancer Research Foundation of Alabama and the National Institutes of Health. N.D., J.C., and R.Nu report current employment with AstraZeneca. P.T. reports advisory/consultancy roles for AstraZeneca, Daiichi Sankyo and Lilly. S.M.T. reports consulting or advisory roles for Novartis, Pfizer, Merck, Eli Lilly, AstraZeneca, Genentech/Roche, Eisai, Sanofi, Bristol Myers Squibb, Seattle Genetics, CytomX Therapeutics, Daiichi Sankyo, Gilead, Onc-Xerna, Zymeworks, Zentalis, Blueprint Medicines, Reveal Genomics, ARC Therapeutics, Infinity Therapeutics, Sumitovant Biopharma, Zetagen, Umoja Biopharma, Artios Pharma, Menarini/Stemline, Aadi Bio, Bayer, Incyte Corp, Jazz Pharmaceuticals, Natera, Tango Therapeutics, Systimmune, eFFECTOR, and Hengrui USA; research funding from Genentech/Roche, Merck, Exelixis, Pfizer, Lilly, Novartis, Bristol Myers Squibb, Eisai, AstraZeneca, Gilead, NanoString Technologies, Seattle Genetics, and OncoPep; and travel support from Eli Lilly, Sanofi, Gilead, and Pfizer. S.M.S. reports serving on advisory boards with honorarium at AstraZeneca, Daiichi-Sankyo, Aventis, Silverback Therapeutics,

Genentech/Roche, Merck, Biotheranostics, Natera, Lilly, Molecular Templates and Exact Sciences; institutional research funding from Kailos Genetics and Genentech/Roche; third party in-kind writing from Genentech/Roche and AstraZeneca; and stock and stock options from Seagen. P.P. reports consulting for BOLT Therapeutics, AbbVie and Perthera; and serving as an unpaid steering committee member of a clinical trial for Seagen. H.A.P. reports institutional research funding from Puma Biotechnology and serving on the advisory board of Illumina. C.I. reports consulting for Genentech, PUMA, Seattle Genetics, Astra-Zeneca, Novartis, Pfizer, ESAI, Sanofi, ION and Gilead; royalties from Wolters Kluwer (UptoDate) and McGraw Hill (Goodman and Gillman); institutional research support from Tesaro/GSK, Seattle Genetics, Pfizer, AstraZeneca, Bristol Myers Squibb, Genentech and Novartis; and serving as a medical director for the Side-Out Foundation. The remaining authors declare no competing interests.

## Additional information

Filipa Lynce ⓘ [1,2,3,18] ✉, Candace Mainor ⓘ [4,18], Renee N. Donahue ⓘ [5,18], Xue Geng ⓘ [6], Greg Jones ⓘ [7], Ilana Schlam ⓘ [8,9], Hongkun Wang [6], Nicole J. Toney [5], Caroline Jochems ⓘ [5], Jeffrey Schlom ⓘ [5], Jay Zeck ⓘ [4], Christopher Gallagher ⓘ [8], Rita Nanda ⓘ [10], Deena Graham ⓘ [11], Erica M. Stringer-Reasor [12], Neelima Denduluri ⓘ [13], Julie Collins ⓘ [4,17], Ami Chitalia ⓘ [8], Shruti Tiwari [8], Raquel Nunes ⓘ [14,17], Rebecca Kaltman [15], Katia Khoury ⓘ [12], Margaret Gatti-Mays [16], Paolo Tarantino ⓘ [1,3], Sara M. Tolaney ⓘ [1,2,3], Sandra M. Swain ⓘ [6], Paula Pohlmann [4], Heather A. Parsons ⓘ [1,2,3,19] & Claudine Isaacs ⓘ [6,19]

[1]Division of Medical Oncology, Dana-Farber Cancer Institute, Boston, MA, USA. [2]Breast Oncology Program, Dana-Farber Brigham Cancer Center, Boston, MA, USA. [3]Harvard Medical School, Boston, MA, USA. [4]MedStar Georgetown University Hospital, Washington, DC, USA. [5]Center for Immuno-Oncology, Center for Cancer Research, National Cancer Institute, National Institutes of Health, Bethesda, MD, USA. [6]Georgetown University, Washington, DC, USA. [7]NeoGenomics, Durham, NC, USA. [8]MedStar Washington Hospital Center, Washington, DC, USA. [9]Tufts Medical Center, Boston, MA, USA. [10]University of

Chicago, Chicago, IL, USA. [11]Hackensack University Medical Center, Hackensack, NJ, USA. [12]University of Alabama at Birmingham, Birmingham, AL, USA. [13]AstraZeneca, Arlington, VA, USA. [14]Johns Hopkins Sidney Kimmel Cancer Center, Baltimore, MD, USA. [15]Inova, Fairfax, VA, USA. [16]The Ohio State University, Columbus, OH, USA. [17]Present address: AstraZeneca, Arlington, VA, USA. [18]These authors contributed equally: Filipa Lynce, Candace Mainor, Renee N. Donahue. [19]These authors jointly supervised this work: Heather A. Parsons, Claudine Isaacs. ✉e-mail: Filipa_Lynce@dfci.harvard.edu

