## [Peer Review File · Nature Communications]

Adjuvant nivolumab, capecitabine or the combination in patients with residual triple-negative breast cancer: the OXEL randomized phase II studyREVIEWER COMMENTS

Reviewer #1 (Remarks to the Author): with expertise in biostatistics, clinical trial study design

Statistical analysis for primary, secondary, and exploratory endpoints was provided appropriately according to the protocol. Sample size was presented clearly in each arm including tissue samples in the Consort Diagram.

Statistical comments:

1. There is no patient baseline demographics data for each Arm in Table 1. The information could help assess whether the randomization process minimized treatment assignment bias.
2. A plot of Capecitabine dose intensity would be helpful for comparison of Arm B and C in addition to Supplementary Table 1.
3. Evaluation of the primary endpoint, peripheral immuno-score, showed reduction in Arm B and increase in Arm A+B at week 6 (Fig 1B). Pairwise comparison also showed arm difference in term of the score change (Fig 1C). However, it is unclear how each drug contributed to the peripheral immuno-score. ANOVA or linear model may be useful to delineate drug effect by including each drug and the interaction term in the model. Did the same pattern hold at week 12?
4. Fig 1 D-F and Supplemental Table 4 showed the changes in peripheral immune profile induced after 6 and 12 weeks of therapy. Most significant biomarkers at week 6 were not significant at week 12. Even the ones with significant change at week 12 (e.g., Nk-DN in Arm C and CD4 naïve in Arm B) had reduced expression (except CD8 CD73+). Does it mean the change is temporary? Also, there were more changes in Arm B compared to Arm A and C. Any explanation? Did capecitabine reduce immune functions? The main text reporting Fig 1D-F should indicate its related treatment Arm to avoid confusion.
5. Survival analysis seems to indicate improved iDFS in Arm C (Fig 2A and Supplementary Table 5). In contrast, the peripheral immuno-score showed reduction in Arm B and increase in Arm A and C. Was there association of the peripheral immuno-score (baseline and the change) with survival outcomes? Any explanation to link both results in terms of drug effect.

6. Toxicity analysis showed (a) 33% patients (5/15) in Arm C experienced drug-related grade 3 toxicity compared to Arm A (n=0) with $p=0.057$, (b) rate of drug-related adverse events of any grade: (14.6%) reported in Arm A, 74 (37.4%) in Arm B, and 95 (48.0%) in Arm C, and (c) rate of at least one drug-related adverse event: 73.3% (11/15) in Arm A and 93.3% (14/15) in Arm B and C. Any explanation why Arm B and C had higher toxicity rate. Did the combination of nivolumab + capecitabine generate synergetic toxicity effect?
7. Was treatment arm associated with recurrence rate? Analysis of immune biomarkers for association of disease recurrence was analyzed at baseline and week 6 and 12 in each arm. While there were a few significant biomarkers in each arm, most were not overlapped between arms and between each time point. Such exploratory analysis makes difficult to untangle the relationship of these biomarkers and drug/treatment in terms of recurrence. Also, false discovery concern should be addressed given multiple testing issue (i.e., p value adjustment).
8. While ctDNA showed negative association with survival outcome, did treatment arms differentially affect the association (e.g., did patients with ctDNA+ have improved survival in Arm C vs. Arm B).

Reviewer #2 (Remarks to the Author): with expertise in breast cancer, clinical, immunotherapy

This is a phase II study of a highly relevant question in the TNBC RD setting. Unfortunately there was no IO given in the early stage setting, and the number of patients per arm is small. However, the KM curves are provocative and the correlative analysis is nice. It is very interesting that the CAPE patients have a reduction in the T cell proliferation, yet still appear to do better.

Comments

Please explain clearly what is the peripheral immunoscore. Provide all components of score in the supplementary and the change from 6 weeks to landmark, as well as at baseline (ie at diagnosis) where possible. Would be useful to the reader in the text to mention if the study achieved its primary endpoint. It clearly was not powered for a survival endpoint and this should also be mentioned.

Please provide PDL1 IHC status as well as TIL status (on the H&E) on the patients, at baseline as well as RD and include in a Table 1 of all the baseline and RD yp tumor stage characteristics by treatment arm given the survival differences. How does TIL – could use a 20% cutoff as previously published correlate with immunoscores?

Could be nice to provide a univariate and multivariate analysis with all relevant significant variables including immunoscore, treatment arm, ctDNA status and PDL1/TIL if significant in the univariate.

I would include all PMBC data changes in the supplementary by treatment arm. Some of the significant data presented in Fig 3 are very small % on the y axis so not sure of the relevance? Can authors provide some context with regards to this? Were absolute numbers looked at, or only frequencies?

Could provide ctDNA data by treatment arm as well in the Supp.

Did the gBRCA patients have any different immune profiles?

Discussion- mention of course that the utility of adjuvant IO is unclear if the patients receive IO in the neoadjuvant setting (timely given SWOG melanoma study recently published in the NEJM). It is likely that the neoadjuvant part is essential only.

Reviewer #3 (Remarks to the Author): with expertise in breast cancer, clinical, immunotherapy

This manuscript shows the result of a phase II study evaluating the immunological effects of adjuvant nivolumab +/- capecitabine compared to capecitabine alone in patients with non-pCR TNBC after chemotherapy without pembrolizumab. This is an important and interesting topic.

The results and conclusions are straightforward.

I have some minor comments:

1 The definition of TNBC was as ER \leq 5%, PR \leq 5%. The authors would mention why the cut-off

did not 1% nor 10% of hormone receptor positivity.

2 The baseline immune profile in this study may be highly influenced by radiation after surgery. I suggest the authors to add information about radiotherapy in this study population. I also suggest that the relationship with the baseline immune profile would be evaluated.

3 Authors mentioned the imbalance of ctDNA between each arms, however, they do not mention the imbalance of pathological staging, which is an important information to interpret the result of prognosis. I suggest to add the information of p-stages in each arms.

REVIEWER COMMENTS

Reviewer #1 (Remarks to the Author): with expertise in biostatistics, clinical trial study design

Statistical analysis for primary, secondary, and exploratory endpoints was provided appropriately according to the protocol. Sample size was presently clearly in each arm including tissue samples in the Consort Diagram.

Statistical comments:

1. There is no patient baseline demographics data for each Arm in Table 1. The information could help assess whether the randomization process minimized treatment assignment bias.

We thank the Reviewer for this suggestion. We added patient baseline demographics for each Arm to Table 1.

2. A plot of Capecitabine dose intensity would be helpful for comparison of Arm B and C in addition to Supplementary Table 1.

We thank the Reviewer for this suggestion. We have now included a boxplot of capecitabine relative dose intensity as **Supplementary Figure 1**.

3. Evaluation of the primary endpoint, peripheral immuno-score, showed reduction in Arm B and increase in Arm A+B at week 6 (Fig 1B). Pairwise comparison also showed arm difference in term of the score change (Fig 1C). However, it is unclear how each drug contributed to the peripheral immunoscore. ANOVA or linear model may be useful to delineate drug effect by including each drug and the interaction term in the model. Did the same pattern hold at week 12?

We thank the reviewer for raising these important questions. To address these comments, we have modified the results section to describe the effect of each individual agent more clearly on the peripheral immunoscore, by describing the effects on the immunoscore in Arm A and B, where these agents were given as monotherapies. In addition, we have included the requested data (Supplemental Figure 1) on the peripheral immunoscore at week 12.

On page 7 of the tracked edits manuscript we have added the following information:

“We evaluated if the calculation of a peripheral immunoscore (immunoscore #1) based on the frequency of specific immune subsets at landmark, 6 and 12 weeks, for which a biologic function has previously been reported⁵⁵⁻⁶⁴, could identify immunologic changes that were unique to each treatment arm (Fig. 1A, Supplementary Table 3). Immunoscore #1, which is reflective of enhanced immune function, was not significantly altered after 6 weeks in patients enrolled in Arm A (who received nivolumab alone, $p=0.217$), or Arm C (who received nivolumab plus chemotherapy, $p=0.100$), but was significantly reduced ($p=0.005$) in patients in Arm B who received chemotherapy alone (Fig. 1B). Significant increases in immunoscore #1 were noted at

6 weeks in analyses combining patients in Arms A and C (who received immunotherapy, $p=0.040$). Only the reduction in the peripheral immunoscore in Arm B was maintained at 12 weeks (Supplementary Fig. 2A). These comparisons were analyzed using a Wilcoxon signed rank test. A comparison of the % change in peripheral immunoscore #1 at 6 and 12 weeks vs landmark further highlights the statistical difference in the immunologic effects of chemotherapy and immunotherapy (Fig. 1C, Supplementary Fig. 2B).”

4. Fig 1 D-F and Supplemental Table 4 showed the changes in peripheral immune profile induced after 6 and 12 weeks of therapy. Most significant biomarkers at week 6 were not significant at week 12. Even the ones with significant change at week 12 (e.g., Nk-DN in Arm C and CD4 naïve in Arm B) had reduced expression (except CD8 CD73+). Does it mean the change is temporary? Also, there were more changes in Arm B compared to Arm A and C. Any explanation? Did capecitabine reduce immune functions? The main text reporting Fig 1D-F should indicate its related treatment Arm to avoid confusion.

We thank the Reviewer for asking these important questions. Both transient and sustained immune changes were observed. We have edited the results section to describe the duration of each individual immune change more clearly, as well as to indicate the treatment arm. We also clarified that Fig 1D-F is related to specific treatment arms. The following is now included on pages 7-8 of the tracked edits manuscript:

“After 6 weeks, patients receiving capecitabine (Arm B) had decreases in proliferative CD8⁺ T cells (ki67⁺, p=0.019), and increases in naïve CD4⁺ T cells (p=0.011), CD8⁺ T cells that express CD73, a checkpoint involved in adenosine metabolism (p=0.020), and NK cells that express the adhesion molecule CD226 (p=0.038) (Fig. 1D). The reduction in ki67⁺ CD8⁺ T cells and increases in naïve CD4⁺ T cells and CD73⁺ CD8⁺ T cells were maintained at 12 weeks. In contrast, patients receiving nivolumab (Arm A) had a transient reduction at 6 weeks in double negative (DN, CD56^{dim} CD16⁻) NK cells (p=0.025) (Fig. 1E), a refined NK subset recently reported to be non-cytolytic and exhausted. Finally, patients receiving the combination therapy (Arm C) had transient increases at 6 weeks in conventional dendritic cells (cDC) (p=0.021), a subset involved in antigen presentation, and proliferative effector memory (EM) CD4⁺ T cells that express ki67 (p=0.037), while DN NK cells were increased after therapy both at 6 and 12 weeks (p=0.033) (Fig. 1F).”

Based on the Reviewer’s very interesting question about the potential immune suppressive effects of capecitabine, we added the following text to the discussion (pages 18-19 of the tracked edits manuscript):

“In general, more extensive changes in peripheral subsets were seen in patients who received capecitabine, either alone or in combination with nivolumab. With capecitabine monotherapy, we observed notable reductions in proliferative CD8⁺ T cells, an immune inhibitory effect, and increases in both CD73⁺ CD8⁺ T cells and CD226⁺ NK cells, which could have variable implications for anti-cancer immunity. In pre-clinical studies, CD73 has been shown to restrict the cytotoxic anti-tumor activity of CD8⁺ T cells⁸⁴, while CD226, an adhesion molecule and

activating receptor has been shown to be important for NK cell anti-tumor activity in vitro⁸⁵ and in patients with cancer⁸⁶. Patients receiving nivolumab plus capecitabine had increases in double negative NK cells, cDCs which are involved in antigen presentation, and proliferative effector memory CD4⁺ T cells subsets. These findings suggest that some of the immune inhibitory actions of capecitabine may have been alleviated by the addition of nivolumab to capecitabine. In a prior study, neoadjuvant chemotherapy caused a transient increase in NK cells, NKT cells, and non-classical monocytes in breast cancer patients⁸⁰. Our results further support that chemotherapy in combination with an ICI causes extensive changes in the peripheral immune system that may enhance anti-tumor activity.”

5. Survival analysis seems to indicate improved iDFS in Arm C (Fig 2A and Supplementary Table 5). In contrast, the peripheral immuno-score showed reduction in Arm B and increase in Arm A and C. Was there association of the peripheral immuno-score (baseline and the change) with survival outcomes? Any explanation to link both results in terms of drug effect.

Thank you for raising this question. The immunoscore in Figure 1, now denoted immunoscore #1, that increases after immunotherapy (Arms A and C combined) and decreases after chemotherapy alone (Arm B), does not associate with disease recurrence or the interval of disease free survival (iDFS). However, the landmark immunoscore in Figure 4, now denoted immunoscore #2, does associate with disease recurrence in certain arms. As requested, we have now investigated whether the immunoscore #2 at landmark associates with iDFS. We have

included new data in Figure 4C and added the following text to the results section (page 11-12 of the tracked edits manuscript):

“To further interrogate the potential value of analyzing peripheral immune cell subsets, we determined if the calculation of another peripheral immunoscore (immunoscore #2), calculated with different refined immune cell subsets than immunoscore 1, and based only on pre-therapy frequencies of specific immune subsets, would be of prognostic value in determining which patients in the current study may most likely benefit from therapy. Immunoscore #2, which is based on specific refined immune cell subsets at landmark (Fig. 4A, Supplementary Table 9) for which a biologic function has been previously reported^{56,66-73} was significantly associated with recurrence in patients receiving nivolumab alone (Arm A, $p=0.005$) or nivolumab +/- capecitabine (Arms A and C combined, $p=0.040$), but showed no association with recurrence in patients receiving capecitabine alone (Arm B, $p=0.576$) (Fig. 4B). Moreover, patients in Arm A (receiving nivolumab, $p=0.0003$) and Arms A and C combined (receiving nivolumab +/- capecitabine, $p=0.0085$) with a landmark immunoscore #2 above the median (> 11) had a longer iDFS compared to patients with an immunoscore #2 at or below the median in these arms (Fig. 4C). In contrast, landmark immunoscore #2 in Arm B (where patients received chemotherapy alone) was not associated with iDFS”

6. Toxicity analysis showed (a) 33% patients (5/15) in Arm C experienced drug-related grade 3 toxicity compared to Arm A (n=0) with $p=0.057$, (b) rate of drug-related adverse events of any grade: (14.6%) reported in Arm A, 74 (37.4%) in Arm B, and 95 (48.0%) in Arm C, and (c) rate of at least one drug-related adverse event: 73.3% (11/15) in Arm A and

93.3% (14/15) in Arm B and C. Any explanation why Arm B and C had higher toxicity rate. Did the combination of nivolumab + capecitabine generate synergetic toxicity effect?

Thank you for bringing up this question. The higher toxicity rate of chemotherapy as monotherapy (Arm B) compared to immunotherapy monotherapy (Arm A) has been previously described. The KEYNOTE-119 trial was a randomized open label phase III study of pembrolizumab monotherapy vs. single agent chemotherapy in 622 patients with previously treated metastatic TNBC. In this study, pembrolizumab was associated with less grade 3-5 treatment related adverse events when compared to chemotherapy (14% vs. 36%) (Winer E et al. Lancet Oncol 2021). The following was added to page 17 of the tracked edits manuscript:

“As previously reported, checkpoint inhibitor given as monotherapy was better tolerated when compared to single agent chemotherapy.”

The findings related to Arm C are thought provoking and it is certainly plausible that the combination of nivolumab plus capecitabine generated synergistic toxicity effects. In the randomized phase III CHECKMATE 649 trial in patients with GI adenocarcinoma, the rate of grade 3-4 treatment-related adverse events was higher among patients treated with nivolumab plus capecitabine (59%) versus capecitabine alone (44%) (Janjigian et al. The Lancet 2021). We added the following to the Discussion (page 17 of the tracked edits manuscript):

“Although no new safety signals were identified in the combination arm, the incidence of drug-related adverse events, including grade 3, was higher in the combination arm, suggesting there

may be some degree of synergistic toxicity between nivolumab and capecitabine. A phase II single arm study of capecitabine and pembrolizumab in patients with HER2 negative advanced breast cancer reported that the combination was well tolerated, and most observed adverse events were low grade and consistent with what would be expected with capecitabine monotherapy. However, our results are aligned with what was observed in the CHECKMATE 649 trial, in which patients with treatment-naïve, HER2-negative, unresectable gastric, gastro-esophageal, or esophageal adenocarcinoma were randomized to receive nivolumab plus chemotherapy (capecitabine and oxaliplatin every 3 weeks or leucovorin, fluorouracil, and oxaliplatin every 2 weeks) or chemotherapy alone. The rate of grade 3-4 treatment-related adverse events was 59% among patients who received nivolumab plus chemotherapy compared to 44% among patients who received chemotherapy alone. In the present study, there were no grade 4 or grade 5 toxicities observed and importantly, there was no increase in irAEs in the combination arm.”

7. Was treatment arm associated with recurrence rate? Analysis of immune biomarkers for association of disease recurrence was analyzed at baseline and week 6 and 12 in each arm. While there were a few significant biomarkers in each arm, most were not overlapped between arms and between each time point. Such exploratory analysis makes difficult to untangle the relationship of these biomarkers and drug/treatment in terms of recurrence. Also, false discovery concern should be addressed given multiple testing issue (i.e., p value adjustment).

We thank the Reviewer for this suggestion. The treatment arm was not associated with iDFS or OS (page 8 of the tracked edits manuscript). In order to avoid concerns with multiple category testing using different immune biomarkers, the peripheral immunoscore #2 was dichotomized for iDFS analyses into “above the baseline median (>11)” vs. “equal or less than the baseline median (≤ 11)”.

We have added the following the results section on page 11-12 of the tracked edits manuscript using this dichotomy:

“...patients in Arm A (receiving nivolumab, $p=0.0003$) and Arms A and C combined (receiving nivolumab +/- capecitabine, $p=0.0085$) with a landmark immunoscore #2 above the median (> 11) had a longer iDFS compared to patients with an immunoscore #2 at or below the median in these arms (Fig. 4C). In contrast, landmark immunoscore #2 in Arm B (where patients received chemotherapy alone) was not associated with iDFS.”

In terms of false discovery, since the analyses of the different immune biomarkers are exploratory in nature, p-value is often not adjusted. We have added to the discussion about the limitation of our findings related to the analysis of immune biomarkers given the small sample size and exploratory nature of the analysis (page 22 of the tracked edits manuscript).

“Moreover, the study was not statistically powered to assess differences in survival outcomes between treatment arms or different landmark immune subsets. Thus, findings from this study are exploratory and should be interpreted as hypothesis-generating.”

8. While ctDNA showed negative association with survival outcome, did treatment arms differentially affect the association (e.g., did patients with ctDNA+ have improved survival in Arm C vs. Arm B).

We thank the Reviewer for this question. We ran our analysis again, and the interaction between baseline ctDNA and arm assigned (either each arm separately or arm A+C vs. Arm B) didn't show significance from neither the multivariate analysis for iDFS nor the multivariate analysis for OS, meaning that the effect of baseline ctDNA on iDFS or OS was not different among arms.

Reviewer #2 (Remarks to the Author): with expertise in breast cancer, clinical, immunotherapy

This is a phase II study of a highly relevant question in the TNBC RD setting. Unfortunately there was no IO given in the early stage setting, and the number of patients per arm is small. However, the KM curves are provocative, and the correlative analysis is nice. It is very interesting that the CAPE patients have a reduction in the T cell proliferation, yet still appear to do better.

Comments

1. Please explain clearly what is the peripheral immunoscore. Provide all components of score in the supplementary and the change from 6 weeks to landmark, as well as at baseline (ie at diagnosis) where possible. Would be useful to the reader in the text to mention if the study achieved its primary endpoint. It clearly was not powered for a survival endpoint and this should also be mentioned.

Thank you very much for this important feedback. To address this comment, we have added further details to the methods on how the immunoscores were developed and referenced **Supplementary Table 3** (containing the refined subsets included in the development of the peripheral immunoscore #1 at landmark, 6 and 12 weeks that was assessed for changes with therapy), and **Supplementary Table 9** (containing the refined subsets included in the development of a peripheral immunoscore #2 at landmark that was associated with disease recurrence).

The following was added to the Methods section (page 26 of the tracked edits manuscript):

“Immune subsets were calculated as a % of PBMC and sorted by frequency. Points were assigned to each subset in a given patient based on tertile distribution. For subsets with an expected positive effect on anti-tumor immunity zero (0) points were assigned to the low bin, one (1) point for the middle bin, and two (2) points if in the high bin. For subsets with an expected negative effect on anti-tumor immunity, zero (0) points were assigned to the high bin, one (1) point for the middle bin, and two (2) points if in the low bin. The peripheral immunoscore for a

given patient was the sum of points assigned to the individual PBMC subsets that were included within the immunoscore.”

In the **Discussion**, we had stated that the study met its primary endpoint, a significant increase in the peripheral immunoscore in patients treated with nivolumab alone or nivolumab plus capecitabine compared to capecitabine alone (pages 16-17 of the tracked edits manuscript). In order to provide more clarification to the reader, we moved this sentence to the beginning of the discussion. We also added that this study was not powered for a survival endpoint. The beginning of the Discussion now reads as follows, with what is underlined representing added information:

“This randomized phase II study was designed to evaluate the role of adjuvant nivolumab, capecitabine or the combination for the treatment of patients with early-stage TNBC with residual invasive disease after the completion of neoadjuvant chemotherapy. The study met the pre-specified primary endpoint, with patients treated with immunotherapy containing regimens (arms A and C) experiencing a greater increase at week 6 versus baseline in a peripheral immunoscore (immunoscore #1) compared to patients treated with chemotherapy alone (Arm B). The combination regimen was associated with a numerical improvement in median iDFS and OS compared to nivolumab or capecitabine monotherapy, although the difference did not reach statistical significance and this study was not powered for survival endpoints.”

We have also amended the limitations section of the Discussion to state that the study was not statistically powered to assess differences in survival outcomes between arms (page 22 of the

tracked edits manuscript):

“Moreover, the study was not statistically powered to assess differences in survival outcomes between treatment arms.”

2. Please provide PDL1 IHC status as well as TIL status (on the H&E) on the patients, at baseline as well as RD and include in a Table 1 of all the baseline and RD yp tumor stage characteristics by treatment arm given the survival differences. How does TIL – could use a 20% cutoff as previously published correlate with immunoscores?

We thank the Reviewer for these suggestions. Unfortunately, PDL1 IHC status and TILs were not available on these specimens. Many of the samples were exhausted after performing whole exome sequencing to identify the unique mutations used to detect ctDNA in the blood. The limited number of samples that were still available wouldn't provide meaningful information.

3. Could be nice to provide a univariate and multivariate analysis with all relevant significant variables including immunoscore, treatment arm, ctDNA status and PDL1/TIL if significant in the univariate.

We thank the Reviewer for this suggestion. We performed univariate analysis of iDFS and OS with the following variables: landmark peripheral immunoscore #1 and #2 (dichotomized peripheral immunoscore #1 into "above the baseline median (>10)" vs. "equal or less than the baseline median (<=10)", and dichotomized peripheral immunoscore #2 into "above the baseline

median (>11)" vs. "equal or less than the baseline median (<=11)", treatment arm and ctDNA status. Only landmark ctDNA status was significantly associated with iDFS or OS in all patients combined; landmark peripheral immunoscore #2 was associated with iDFS in certain arms. Therefore, we only performed multivariate analysis for iDFS to further evaluate the effects of landmark ctDNA, landmark peripheral immunoscore #2, and treatment arms. Based on the multivariate Cox proportional hazard model, patients who were ctDNA-positive at landmark had significantly worse iDFS compared to patients who were ctDNA-negative: hazard ratio (HR) 50.70 (95% CI: 6.52 – 393.98, $p < 0.001$). Compared to the patients with a landmark immunoscore #2 equal or below the median (<=11), those with a landmark immunoscore #2 above the median had significantly better iDFS: HR 0.064 (95% CI: 0.009 – 0.462, $p = 0.006$). When analyzed by treatment arm, patients with landmark immunoscore #2 equal or below the median treated in Arm C experienced significantly improved iDFS compared to those treatment in Arm A (HR 0.027, 95% CI: 0.002 - 0.35, $p = 0.0058$) but there were no differences between Arm B and Arm A. This was added to the tracked edits manuscript on page 15.

4. I would include all PMBC data changes in the supplementary by treatment arm. Some of the significant data presented in Fig 3 are very small % on the y axis so not sure of the relevance? Can authors provide some context with regards to this? Were absolute numbers looked at, or only frequencies?

We thank the Reviewer for these suggestions. As requested, we have now added additional supplemental data to show the significant PMBC changes that are seen within a given arm across

all arms of the study. This data is in Supplementary Tables 5, 7, and 8 and is referenced in the Results section. Absolute numbers (e.g. # of cells/ul of blood) were unable to be evaluated as immune subset analyses were performed using cryopreserved blood. We have also clarified in the Methods that peripheral immune subsets were calculated as a percentage of total PBMCs, and that very rare subsets were excluded from immune subset analyses, in an effort to focus on potentially more biologically relevant subsets. The following underlined text was added to the Methods section (pages 25-26 of the tracked edits manuscript):

“The frequency of all immune subsets was calculated as a percentage of total PBMCs to eliminate any bias that might occur in the smaller populations with fluctuations in parental leukocyte subpopulations... Peripheral immune subsets with changes following therapy were defined as those with a $p < 0.05$, $\geq 50\%$ of patients having a $> 25\%$ change, and difference in medians of pre- vs post-therapy $> 0.05\%$ of PBMCs. Immune subsets with median values comprising $< 0.01\%$ of total PBMCs were excluded from analyses in an effort to focus on more potentially biologically relevant immune subsets.”

5. Could provide ctDNA data by treatment arm as well in the Supp.

We thank the Reviewer for this suggestion. As requested, we added a Supplementary Table 10 with the information regarding ctDNA dynamics by treatment arm.

6. Did the gBRCA patients have any different immune profiles?

Thank you for this important question. As requested, we have gone back to evaluate the landmark immune profile of patients with known germline deleterious *BRCA1/2* mutations and compared them with patients without known deleterious mutations. As reported in Table 1, only 3 patients in this study had germline BRCA mutations. We have added the following to the results (page 12 of the tracked edits manuscript) and included this data in Supplemental Figure 4C.

“We also investigated landmark differences in patients with germline BRCA1 or BRCA2 (gBRCA1/2) mutations compared to patients without deleterious germline mutations. For this analysis, all patients were similarly combined due to the small number of patients with BRCA1/2 mutations enrolled (2/11 in Arm A, 1/11 in Arm B, and 0/12 in Arm C). While we found a higher percent of CD73⁺ CD8⁺ T cells (p=0.005) in patients with gBRCA1/2 mutations, no other landmark differences were observed (Supplementary Figure 4C).”

7. Discussion- mention of course that the utility of adjuvant IO is unclear if the patients receive IO in the neoadjuvant setting (timely given SWOG melanoma study recently published in the NEJM). It is likely that the neoadjuvant part is essential only.

We thank the Reviewer for this suggestion. We have cited the positive results of the SWOG S1801 trial (Patel et al. NEJM 2023) and the KEYNOTE-522 trial in the Discussion. The

following has been added to the discussion (page 22 of the tracked edits manuscript):

“Given the findings of the KEYNOTE-522 trial, as well as the positive results of SWOG S1801 trial, which compared neoadjuvant pembrolizumab followed by adjuvant pembrolizumab to adjuvant pembrolizumab alone among patients with stage IIIB to IVC melanoma, it is possible that most of the benefit achieved with checkpoint inhibitors may be obtained in the neoadjuvant setting.”

Reviewer #3 (Remarks to the Author): with expertise in breast cancer, clinical, immunotherapy

This manuscript shows the result of a phase II study evaluating the immunological effects of adjuvant nivolumab +/- capecitabine compared to capecitabine alone in patients with non-pCR TNBC after chemotherapy without pembrolizumab. This is an important and interesting topic. The results and conclusions are straightforward.

I have some minor comments:

1. The definition of TNBC was as ER \leq 5%, PR \leq 5%. The authors would mention why the cut-off did not 1% nor 10% of hormone receptor positivity.

Thank you for this important question. We defined the eligibility criteria for this trial in 2018, before the ASCO/CAP guidelines defined the category of ER low positive breast cancers as tumors with 1% to 10% of tumor nuclei staining positive for ER (Allison KH et al. J Clin Oncol 2020). At the time we designed our trial, we followed similar eligibility criteria of other larger studies mainly focused on basal like tumors (TBCRC 043, SWOG S1418) with the assumption that a percentage of borderline low HR-positive tumors will be basal tumors by intrinsic subtyping and can benefit from standard treatments usually offered to triple-negative breast cancers, we decided to include some of these patients in this trial by broadening the ER/PR positivity inclusion criteria to 5%. In both TBCRC048 and SWOG S1418, patients with weekly ER or PR-positive disease, defined as ER and/or PR < 5% by immunohistochemistry, were eligible.

2. The baseline immune profile in this study may be highly influenced by radiation after surgery. I suggest the authors to add information about radiotherapy in this study population. I also suggest that the relationship with the baseline immune profile would be evaluated.

This is an excellent point, and we thank the Reviewer for these suggestions. To address this important point, we have included prior radiotherapy to the patient demographics (Table 1) and modified the results section to include the following (page 6 of the tracked edits manuscript):

“Most patients (76%) had received prior adjuvant radiotherapy.”

Furthermore, we have included additional data in Supplemental Figure 4A and B on the effect of prior radiotherapy on peripheral immune subsets and added the following to the results (page 12 of the tracked edits manuscript):

“We next investigated potential factors that may contribute to landmark variation in the immune profile of patients. We interrogated whether there were immunologic differences among patients based on prior exposure to radiotherapy. For this analysis, all patients were combined due to the limited number of patients who had not received prior radiotherapy in each arm (4/11 in Arm A and Arm B, and 3/12 in Arm C). Compared to patients who didn’t receive radiotherapy, we found that patients who had received prior adjuvant radiotherapy had lower levels of landmark T cells, including total CD4⁺ T cells (p=0.012), PD-L1⁺ CD4⁺ T cells (p=0.016), ICOS⁺ CD4⁺ T cells (p=0.004), central memory (CM) CD4⁺ T cells (p=0.025), and CM CD8⁺ T cells (p=0.022) (Supplementary Fig. 4A), higher levels of landmark monocytes, including total monocytes (p=0.016), PD-L1⁺ monocytes (p=0.016), classical monocytes (p=0.041), PD-L1⁺ classical monocytes (p=0.019), and intermediate monocytes (p=0.016), and higher levels of pDCs (p=0.006), MDSCs (p=0.020), PD-L1⁺ MDSCs (p=0.014), ki67⁺ NK (p=0.047), and NKG2D⁺ immature NK cells (p=0.026) (Supplementary Fig. 4B).”

We have also added the following to the discussion section to discuss the radiotherapy associated immune effects on the immune profile (page 20 of the tracked edits manuscript):

As mentioned above, prior therapy has the potential to induce systemic changes to the immune

system; therefore, in this population where 76% of patients had received prior radiotherapy, we investigated whether differences existed in the landmark immune profile of patients who received versus did not receive prior radiotherapy. We found that patients who received prior adjuvant radiotherapy had decreased levels of total CD4⁺ and refined CD4⁺ T cell subsets and CM CD8⁺ T cells compared to those who didn't receive radiotherapy. Prior studies also report reduction in CD4⁺ T cells following RT in patients with various solid tumors Interestingly, we also observed increases in peripheral immune subsets of monocytes and MDSCs in patients who received prior radiotherapy. Monocytes have been found to be more resistant to radiotherapy than lymphocytes and expansions in MDSCs have also been noted following radiotherapy.”

3. Authors mentioned the imbalance of ctDNA between each arms, however, they do not mention the imbalance of pathological staging, which is an important information to interpret the result of prognosis. I suggest to add the information of p-stages in each arms.

Thank you for this important comment. We added pathological staging information to Table 1. We didn't find any statistical difference in pathological staging among the 3 arms. We added to the Results (page 6):

“Most patients had pathological stage (yp) II. There were no statistical differences in pathological stage among the 3 arms (p=0.36).”

REVIEWERS' COMMENTS

Reviewer #1 (Remarks to the Author):

The authors have adequately addressed statistical concerns.

Reviewer #3 (Remarks to the Author):

The authors well-revised the manuscript in accordance with the reviewers' comments. I think this manuscript is acceptable for publication.

Please note: When reviewing the ctDNA longitudinal data again, we discovered an error in the plotting of one of the patients - UOC-4001 (formerly Supplementary Figure 4, now Supplementary Figure 6). This patient had the first three timepoints plotted as “detected” when properly it should have been “undetected, detected, undetected”. This error has been corrected in the new version of Supplementary Figure 6 that is included with this revision.

RESPONSES TO REVIEWERS' COMMENTS

Reviewer #1 (Remarks to the Author):

The authors have adequately addressed statistical concerns.

We thank the Reviewer for this feedback; we are glad to know that we adequately addressed the statistical concerns.

Reviewer #3 (Remarks to the Author):

The authors well-revised the manuscript in accordance with the reviewers' comments. I think this manuscript is acceptable for publication.

We thank the Reviewer for this positive feedback; we are glad to know that we revised the manuscript sufficiently to address all the Reviewer's concerns.